# GENERATIVE ADVERSARIAL NETWORKS FOR EXTREME LEARNED IMAGE COMPRESSION

## ABSTRACT

We propose a framework for extreme learned image compression based on Generative Adversarial Networks (GANs), obtaining visually pleasing images at significantly lower bitrates than previous methods. This is made possible through our GAN formulation of learned compression combined with a generator/decoder which operates on the full-resolution image and is trained in combination with a multi-scale discriminator. Additionally, if a semantic label map of the original image is available, our method can fully synthesize unimportant regions in the decoded image such as streets and trees from the label map, therefore only requiring the storage of the preserved region and the semantic label map. A user study confirms that for low bitrates, our approach is preferred to state-of-the-art methods, even when they use more than double the bits.

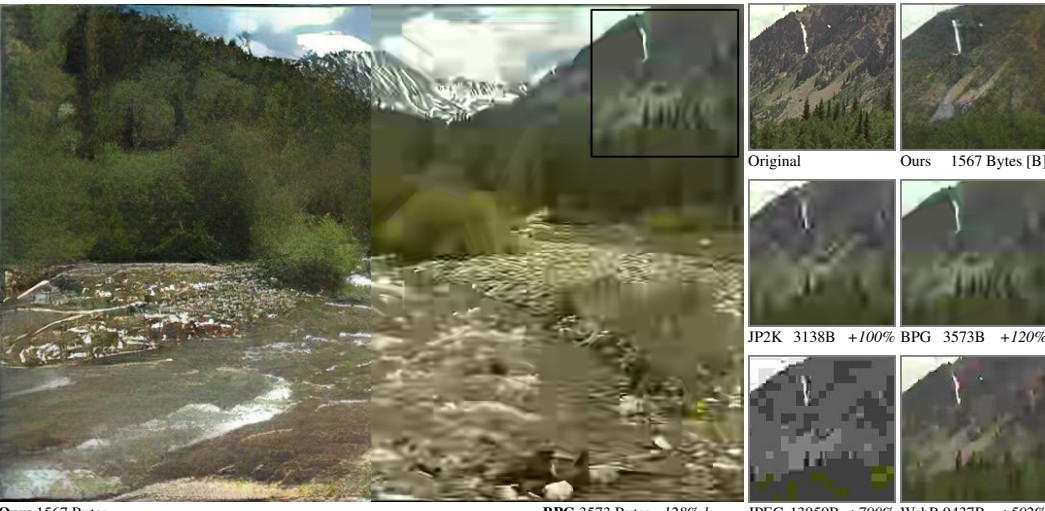

Figure 1: Visual comparison of our result to that obtained by other codecs. Note that even when using more than twice the number of bytes, all other codecs are outperformed by our method visually.

## 1 INTRODUCTION

Image compression systems based on deep neural networks (DNNs), or deep compression systems for short, have become an active area of research recently. These systems (e.g. (Theis et al., 2017; Ballé et al., 2016b; Rippel & Bourdev, 2017; Ballé et al., 2018; Mentzer et al., 2018)) are often competitive with modern engineered codecs such as WebP (WebP), JPEG2000 (Taubman & Marcellin, 2001) and even BPG(Bellard) (the state-of-the-art engineered codec). Besides achieving competitive compression rates on natural images, they can be easily adapted to specific target domains such as stereo or medical images, and promise efficient processing and indexing directly from compressed representations (Torfason et al., 2018). However, deep compression systems are typically optimized for traditional distortion metrics such as peak signal-to-noise ratio (PSNR) or multi-scale structural similarity (MS-SSIM) (Wang et al., 2003). For very low bitrates (below 0.1 bits per pixel (bpp)), where preserving the full image content becomes impossible, these distortion metrics lose significance as they favor pixel-wise preservation of local (high-entropy) structure over preserving texture

and global structure. To further advance deep image compression it is therefore of great importance to develop new training objectives beyond PSNR and MS-SSIM. A promising candidate towards this goal are adversarial losses (Goodfellow et al., 2014) which were shown recently to capture global semantic information and local texture, yielding powerful generators that produce visually appealing high-resolution images from semantic label maps (Isola et al., 2017; Wang et al., 2018).

In this paper, we propose and study a generative adversarial network (GAN)-based framework for extreme image compression, targeting bitrates below 0.1 bpp. We rely on a principled GAN formulation for deep image compression that allows for different degrees of content generation. In contrast to prior works on deep image compression which applied adversarial losses to image patches for artifact suppression (Rippel & Bourdev, 2017; Galteri et al., 2017), generation of texture details (Ledig et al., 2017), or representation learning for thumbnail images (Santurkar et al., 2017), our generator/decoder operates on the full-resolution image and is trained with a multi-scale discriminator (Wang et al., 2018).

We consider two modes of operation (corresponding to unconditional and conditional GANs (Goodfellow et al., 2014; Mirza & Osindero, 2014)), namely

- *generative compression (GC)*, preserving the overall image content while generating structure of different scales such as leaves of trees or windows in the facade of buildings, and
- *selective generative compression (SC)*, completely generating parts of the image from a semantic label map while preserving user-defined regions with a high degree of detail.

We emphasize that GC does not require semantic label maps (neither for training, nor for deployment). A typical use case for GC are bandwidth constrained scenarios, where one wants to preserve the full image as well as possible, while falling back to synthesized content instead of blocky/blurry blobs for regions for which not sufficient bits are available to store the original pixels. SC could be applied in a video call scenario where one wants to fully preserve people in the video stream, but a visually pleasing synthesized background serves the purpose as well as the true background. In the GC operation mode the image is transformed into a bitstream and encoded using arithmetic coding. SC requires a semantic/instance label map of the original image which can be obtained using off-the-shelf semantic/instance segmentation networks, e.g., PSPNet (Zhao et al., 2017) and Mask R-CNN (He et al., 2017), and which is stored as a vector graphic. This amounts to a small, image dimension-independent overhead in terms of coding cost. On the other hand, the size of the compressed image is reduced proportionally to the area which is generated from the semantic label map, typically leading to a significant overall reduction in storage cost.

For GC, a comprehensive user study shows that our compression system yields visually considerably more appealing results than BPG (Bellard) (the current state-of-the-art engineered compression algorithm) and the recently proposed autoencoder-based deep compression (AEDC) system (Mentzer et al., 2018). In particular, our GC models trained for compression of general natural images are preferred to BPG when BPG uses up to 95% and 124% more bits than those produced by our models on the Kodak (Kodak) and RAISE1K (Dang-Nguyen et al., 2015) data set, respectively. When constraining the target domain to the street scene images of the Cityscapes data set (Cordts et al., 2016), the reconstructions of our GC models are preferred to BPG even when the latter uses up to 181% more bits. To the best of our knowledge, these are the first results showing that a deep compression method outperforms BPG on the Kodak data set in a user study—and by large margins. In the SC operation mode, our system seamlessly combines preserved image content with synthesized content, even for regions that cross multiple object boundaries, while faithfully preserving the image semantics. By partially generating image content we achieve bitrate reductions of over 50% without notably degrading image quality.

## 2 RELATED WORK

Deep image compression has recently emerged as an active area of research. The most popular DNN architectures for this task are to date auto-encoders (Theis et al., 2017; Ballé et al., 2016b; Agustsson et al., 2017; Li et al., 2017; Torfason et al., 2018; Minnen et al., 2018) and recurrent neural networks (RNNs) (Toderici et al., 2015; 2016). These DNNs transform the input image into a bit-stream, which is in turn losslessly compressed using entropy coding methods such as Huffman coding or arithmetic coding. To reduce coding rates, many deep compression systems rely on context models

to capture the distribution of the bit stream (Ballé et al., 2016b; Toderici et al., 2016; Li et al., 2017; Rippel & Bourdev, 2017; Mentzer et al., 2018). Common loss functions to measure the distortion between the original and decompressed images are the mean-squared error (MSE) (Theis et al., 2017; Ballé et al., 2016b; Agustsson et al., 2017; Li et al., 2017; Ballé et al., 2018; Torfason et al., 2018), or perceptual metrics such as MS-SSIM (Toderici et al., 2016; Rippel & Bourdev, 2017; Ballé et al., 2018; Mentzer et al., 2018). Some authors rely on advanced techniques including multi-scale decompositions (Rippel & Bourdev, 2017), progressive encoding/decoding strategies (Toderici et al., 2015; 2016), and generalized divisive normalization (GDN) layers (Ballé et al., 2016b;a).

Generative adversarial networks (GANs) (Goodfellow et al., 2014) have emerged as a popular technique for learning generative models for intractable distributions in an unsupervised manner. Despite stability issues (Salimans et al., 2016; Arjovsky & Bottou, 2017; Arjovsky et al., 2017; Mao et al., 2017), they were shown to be capable of generating more realistic and sharper images than prior approaches and to scale to resolutions of $1024 \times 1024$px (Zhang et al., 2017; Karras et al., 2017) for some datasets. Another direction that has shown great progress are conditional GANs (Goodfellow et al., 2014; Mirza & Osindero, 2014), obtaining impressive results for image-to-image translation (Isola et al., 2017; Wang et al., 2018; Zhu et al., 2017; Liu et al., 2017) on various datasets (e.g. maps to satellite images), reaching resolutions as high as $1024 \times 2048$px (Wang et al., 2018).

Arguably the most closely related work to ours is (Rippel & Bourdev, 2017), which uses an adversarial loss term to train a deep compression system. However, this loss term is applied to small image patches and its purpose is to suppress artifacts rather than to generate image content. Furthermore, it uses a non-standard GAN formulation that does not (to the best of our knowledge) have an interpretation in terms of divergences between probability distributions, as in (Goodfellow et al., 2014; Nowozin et al., 2016). We refer to Sec. 6.1 and Appendix A for a more detailed discussion. Santurkar et al. (2017) use a GAN framework to learn a generative model over thumbnail images, which is then used as a decoder for thumbnail image compression. Other works use adversarial training for compression artifact removal (for engineered codecs) (Galteri et al., 2017) and single image super-resolution (Ledig et al., 2017). Finally, related to our SC mode, spatially allocating bitrate based on saliency of image content has a long history in the context of engineered compression algorithms, see, e.g.,, (Stella & Lisin, 2009; Guo & Zhang, 2010; Gupta et al., 2013).

## 3 BACKGROUND

**Generative Adversarial Networks:** Given a data set $\mathcal{X}$, Generative Adversarial Networks (GANs) can learn to approximate its (unknown) distribution $p_{\boldsymbol{x}}$ through a generator $G(\boldsymbol{z})$ that tries to map samples $\boldsymbol{z}$ from a fixed prior distribution $p_{\boldsymbol{z}}$ to the distribution $p_{\boldsymbol{x}}$. The generator $G$ is trained in parallel with a discriminator $D$ by searching (using stochastic gradient descent (SGD)) for a saddle point of a mini-max objective

$$\min_{G} \max_{D} \quad \mathbb{E}[f(D(\boldsymbol{x}))] + \mathbb{E}[g(D(G(\boldsymbol{z})))], \tag{1}$$

where $G$ and $D$ are DNNs and $f$ and $g$ are scalar functions. The original paper (Goodfellow et al., 2014) uses the "Vanilla GAN" objective with $f(y) = \log(y)$ and $g(y) = \log(1-y)$. This corresponds to $G$ minimizing the Jensen-Shannon (JS) Divergence between the (empirical) distribution of $\boldsymbol{x}$ and $G(\boldsymbol{z})$. The JS Divergence is a member of a more generic family of $f$-divergences, and Nowozin et al. (2016) show that for suitable choices of $f$ and $g$, all such divergences can be minimized with (1). In particular, if one uses $f(y) = (y-1)^2$ and $g(y) = y^2$, one obtains the Least-Squares GAN (Mao et al., 2017) (which corresponds to the Pearson $\chi^2$ divergence), which we adopt in this paper. We refer to the divergence minimized over $G$ as

$$\mathcal{L}_{\text{GAN}} := \max_{D} \quad \mathbb{E}[f(D(\boldsymbol{x}))] + \mathbb{E}[g(D(G(\boldsymbol{z})))]. \tag{2}$$

**Conditional Generative Adversarial Networks:** For conditional GANs (cGANs) (Goodfellow et al., 2014; Mirza & Osindero, 2014), each data point $\boldsymbol{x}$ is associated with additional information $\boldsymbol{s}$, where $(\boldsymbol{x}, \boldsymbol{s})$ have an unknown joint distribution $p_{\boldsymbol{x},\boldsymbol{s}}$. We now assume that $\boldsymbol{s}$ is given and that we want to use the GAN to model the conditional distribution $p_{\boldsymbol{x}|\boldsymbol{s}}$. In this case, both the generator $G(\boldsymbol{z}, \boldsymbol{s})$ and discriminator $D(\boldsymbol{z}, \boldsymbol{s})$ have access to the side information $\boldsymbol{s}$, leading to the divergence

$$\mathcal{L}_{\text{cGAN}} := \max_{D} \quad \mathbb{E}[f(D(\boldsymbol{x}, \boldsymbol{s}))] + \mathbb{E}[g(D(G(\boldsymbol{z}, \boldsymbol{s}), \boldsymbol{s}))]. \tag{3}$$

**Deep Image Compression:** To compress an image $x \in \mathcal{X}$, we follow the formulation of (Agustsson et al., 2017; Mentzer et al., 2018) where one learns an encoder $E$, a decoder $G$, and a finite quantizer $q$. The encoder $E$ maps the image to a latent feature map $w$, whose values are then quantized to $L$ levels $\{c_1, \ldots, c_L\} \subset \mathbb{R}$ to obtain a representation $\hat{w} = q(E(x))$ that can be encoded to a bitstream. The decoder then tries to recover the image by forming a reconstruction $\hat{x} = G(\hat{w})$. To be able to backpropagate through the non-differentiable $q$, one can use a differentiable relaxation of $q$, as in (Mentzer et al., 2018).

The average number of bits needed to encode $\hat{w}$ is measured by the entropy $H(\hat{w})$, which can be modeled with a prior (Agustsson et al., 2017) or a conditional probability model (Mentzer et al., 2018). The trade-off between reconstruction quality and bitrate to be optimized is then

$$\mathbb{E}[d(x, \hat{x})] + \beta H(\hat{w}). \tag{4}$$

where $d$ is a loss that measures how perceptually similar $\hat{x}$ is to $x$. Given a differentiable estimator of the entropy $H(\hat{w})$, the weight $\beta$ controls the bitrate of the model (large $\beta$ pushes the bitrate down). However, since the number of dimensions $\dim(\hat{w})$ and the number of levels $L$ are finite, the entropy is bounded by (see, e.g., (Cover & Thomas, 2012))

$$H(\hat{w}) \leq \dim(\hat{w}) \log_2(L). \tag{5}$$

It is therefore also valid to set $\beta = 0$ and control the maximum bitrate through the bound (5) (i.e., adjusting $L$ and/or $\dim(\hat{w})$ through the architecture of $E$). While potentially leading to suboptimal bitrates, this avoids to model the entropy explicitly as a loss term.

## 4 GANs FOR EXTREME IMAGE COMPRESSION

### 4.1 GENERATIVE COMPRESSION

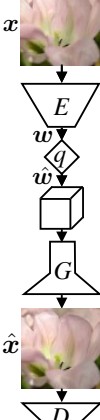

The proposed GAN framework for extreme image compression can be viewed as a combination of (conditional) GANs and learned compression. With an encoder $E$ and quantizer $q$, we encode the image $x$ to a compressed representation $\hat{w} = q(E(x))$. This representation is optionally concatenated with noise $v$ drawn from a fixed prior $p_v$, to form the latent vector $z$. The decoder/generator $G$ then tries to generate an image $\hat{x} = G(z)$ that is consistent with the image distribution $p_x$ while also recovering the specific encoded image $x$ to a certain degree (see inset Fig.). Using $z = [\hat{w}, v]$, this can be expressed by our saddle-point objective for (unconditional) generative compression,

$$\min_{E,G} \max_{D} \quad \mathbb{E}[f(D(x))] + \mathbb{E}[g(D(G(z))] + \lambda \mathbb{E}[d(x, G(z))] + \beta H(\hat{w}), \tag{6}$$

where $\lambda > 0$ balances the distortion term against the GAN loss and entropy terms. Using this formulation, we need to encode a real image, $\hat{w} = E(x)$, to be able to sample from $p_{\hat{w}}$. However, this is not a limitation as our goal is to compress real images and not to generate completely new ones.

Since the last two terms of (6) do not depend on the discriminator $D$, they do not affect its optimization directly. This means that the discriminator still computes the same $f$ divergence $\mathcal{L}_{\text{GAN}}$ as in (2), so we can write (6) as

$$\min_{E,G} \quad \mathcal{L}_{\text{GAN}} + \lambda \mathbb{E}[d(x, G(z))] + \beta H(\hat{w}). \tag{7}$$

We note that equation (6) has completely different dynamics than a normal GAN, because the latent space $z$ contains $\hat{w}$, which stores information about a real image $x$. A crucial ingredient is the bitrate limitation on $H(\hat{w})$. If we allow $\hat{w}$ to contain arbitrarily many bits by setting $\beta = 0$ and letting $L$ and $\dim(\hat{w})$ be large enough, $E$ and $G$ could learn to near-losslessly recover $x$ from $G(z) = G(q(E(x)))$, such that the distortion term would vanish. In this case, the divergence between $p_x$ and $p_{G(z)}$ would also vanish and the GAN loss would have no effect.

By constraining the entropy of $\hat{w}$, $E$ and $G$ will never be able to make $d$ fully vanish. In this case, $E, G$ need to balance the GAN objective $\mathcal{L}_{\text{GAN}}$ and the distortion term $\lambda \mathbb{E}[d(x, G(z))]$, which leads to $G(z)$ on one hand looking "realistic", and on the other hand preserving the original image. For example, if there is a tree for which $E$ cannot afford to store the exact texture (and make $d$ small) $G$ can synthesize it to satisfy $\mathcal{L}_{\text{GAN}}$, instead of showing a blurry green blob.

In the extreme case where the bitrate becomes zero (i.e., $H(\hat{w}) \to 0$, e.g., by setting $\beta = \infty$ or $\dim(\hat{w}) = 0$), $\hat{w}$ becomes deterministic. In this setting, $z$ is random and independent of $x$ (through the $v$ component) and the objective reduces to a standard GAN plus the distortion term, which acts as a regularizer.

We refer to the setting in (6) as *generative compression* (GC), where $E, G$ balance reconstruction and generation automatically over the image. As for the conditional GANs described in Sec. 3, we can easily extend GC to a conditional case. Here, we also consider this setting, where the additional information $s$ for an image $x$ is a semantic label map of the scene, but with a twist: Instead of feeding the semantics to $E, G$ and $D$, we *only give them to the discriminator $D$* during training.[1] This means that *no semantics are needed* to encode or decode images with the trained models (since $E, G$ do not depend on $s$). We refer to this setting as GC ($D^+$).

### 4.2 SELECTIVE GENERATIVE COMPRESSION

For GC and its conditional variant described in the previous section, $E, G$ automatically navigate the trade-off between generation and preservation over the entire image, without any guidance. Here, we consider a different setting, where we guide the network in terms of which regions should be preserved and which regions should be synthesized. We refer to this setting as *selective generative compression* (SC) (an overview of the network structure is given in Fig. 8 in Appendix C).

For simplicity, we consider a binary setting, where we construct a single-channel binary heatmap $m$ of the same spatial dimensions as $\hat{w}$. Regions of zeros correspond to regions that should be fully synthesized, whereas regions of ones correspond to regions that should be preserved. However, since our task is compression, we constrain the fully synthesized regions to have the same semantics $s$ as the original image $x$. We assume the semantics $s$ are separately stored, and thus feed them through a feature extractor $F$ before feeding them to the generator $G$. To guide the network with the semantics, we mask the (pixel-wise) distortion $d$, such that it is only computed over the region to be preserved. Additionally, we zero out the compressed representation $\hat{w}$ in the regions that should be synthesized. Provided that the heatmap $m$ is also stored, we then only encode the entries of $\hat{w}$ corresponding to the preserved regions, greatly reducing the bitrate needed to store it.

At bitrates where $\hat{w}$ is normally much larger than the storage cost for $s$ and $m$ (about 2kB per image when encoded as a vector graphic), this approach can result in large bitrate savings.

We consider two different training modes: Random instance (RI) which randomly selects 25% of the instances in the semantic label map and preserves these, and random box (RB) which picks an image location uniformly at random and preserves a box of random dimensions. While the RI mode is appropriate for most use cases, the RB can create more challenging situations for the generator as it needs to integrate the preserved box seamlessly into the generated content.

## 5 EXPERIMENTS

### 5.1 ARCHITECTURE, LOSSES, AND HYPERPARAMETERS

The architecture for our encoder $E$ and generator $G$ is based on the global generator network proposed in (Wang et al., 2018), which in turn is based on the architecture of (Johnson et al., 2016). We present details in Appendix C.

For the entropy term $\beta H(\hat{w})$, we adopt the simplified approach described in Sec. 3, where we set $\beta = 0$, use $L = 5$ centers $\mathcal{C} = \{-2, 1, 0, 1, 2\}$, and control the bitrate through the upper bound $H(\hat{w}) \le \dim(\hat{w}) \log_2(L)$. For example, for GC, with $C = 2$ channels, we obtain 0.0181bpp.[2] We note that this is an upper bound; the actual entropy of $H(\hat{w})$ is generally smaller, since the learned distribution will neither be uniform nor i.i.d, which would be required for the bound to hold with equality. When encoding the channels of $\hat{w}$ to a bit-stream, we use an arithmetic encoder where frequencies are stored for each channel separately and then encode them in a static (non-adaptive)

---

[1]If we assume $s$ is an unknown function of $x$, another view is that we feed additional features ($s$) to $D$.

[2] $H(\hat{w})/WH \le \frac{WH}{16 \cdot 16} \cdot C \cdot \log_2(L)/WH = 0.0181$bpp, where $W, H$ are the dimensions of the image and 16 is the downsampling factor to the feature map, see Sec. C in the Appendix.

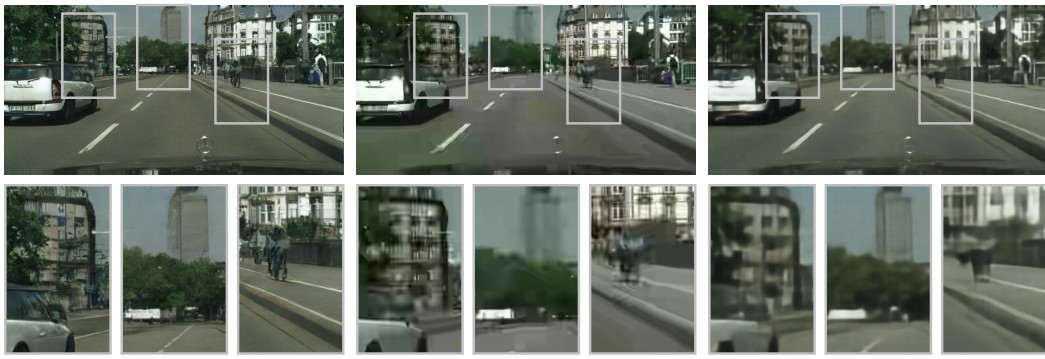

| Ours, 0.035bpp, 21.8dB | BPG, 0.039bpp, 26.0dB | MSE bl., 0.035bpp, 24.0dB |

Figure 2: Visual example of images produced by our GC network with $C = 4$ along with the corresponding results for BPG, and a baseline model with the same architecture ($C = 4$) but trained for MSE only (MSE bl.), on Cityscapes. The reconstruction of our GC network is sharper and has more realistic texture than those of BPG and the MSE baseline, even though the latter two have higher PSNR (indicated in dB for each image) than our GC network. In particular, the MSE baseline produces blurry reconstructions even though it was trained on the Cityscapes data set, demonstrating that domain-specific training alone is not enough to obtain sharp reconstructions at low bitrates.

manner, similar to Agustsson et al. (2017). In our experiments, this leads to $8.8\%$ smaller bitrates compared to the upper bound.

By using a context model and adaptive arithmetic encoding, we could reduce the bitrate further, either in a post processing step (as in (Rippel & Bourdev, 2017; Ballé et al., 2016b)), or jointly during training (as in (Mentzer et al., 2018; Minnen et al., 2018))—which led to $\approx 10\%$ savings in these prior works.

For the distortion term we adopt $d(\boldsymbol{x}, \hat{\boldsymbol{x}}) = \mathrm{MSE}(\boldsymbol{x}, \hat{\boldsymbol{x}})$ with coefficient $\lambda = 10$. Furthermore, we adopt the feature matching and VGG perceptual losses, $\mathcal{L}_{\mathrm{FM}}$ and $\mathcal{L}_{\mathrm{VGG}}$, as proposed in (Wang et al., 2018) with the same weights, which improved the quality for images synthesized from semantic label maps. These losses can be viewed as a part of $d(\boldsymbol{x}, \hat{\boldsymbol{x}})$. However, we do not mask them in SC, since they also help to stabilize the GAN in this operation mode (as in (Wang et al., 2018)). We refer to Appendix D for training details.

## 5.2 Evaluation

**Data sets:** We train GC models (without semantic label maps) for compression of diverse natural images using 188k images from the *Open Images* data set (Krasin et al., 2017) and evaluate them on the widely used Kodak image compression data set (Kodak) as well as 20 randomly selected images from the *RAISE1K* data set (Dang-Nguyen et al., 2015). To investigate the benefits of having a somewhat constrained application domain and semantic information at training time, we also train GC models with semantic label maps on the *Cityscapes* data set (Cordts et al., 2016), using 20 randomly selected images from the validation set for evaluation. To evaluate the proposed SC method (which requires semantic label maps for training and deployment) we again rely on the Cityscapes data set. Cityscapes was previously used to generate images form semantic label maps using GANs (Isola et al., 2017; Zhu et al., 2017).

**Baselines:** We compare our method to the HEVC-based image compression algorithm BPG (Bellard) (in the 4:2:2 chroma format) and to the AEDC network from (Mentzer et al., 2018). BPG is the current state-of-the-art engineered image compression codec and outperforms other recent codecs such as JPEG2000 and WebP on different data sets in terms of PSNR (see, e.g. (Ballé et al., 2018)). We train the AEDC network (with bottleneck depth $C = 4$) on Cityscapes exactly following the procedure in (Mentzer et al., 2018) except that we use early stopping to prevent overfitting (note that Cityscapes is much smaller than the ImageNet dataset used in (Mentzer et al., 2018)). The so-obtained model has a bitrate of 0.07 bpp and gets a slightly better MS-SSIM than BPG at the same bpp on the validation set. To investigate the effect of the GAN term in our total loss, we train a baseline model with an MSE loss only (with the same architecture as GC and the same training parameters, see Sec. D in the Appendix), referred to as "MSE baseline".

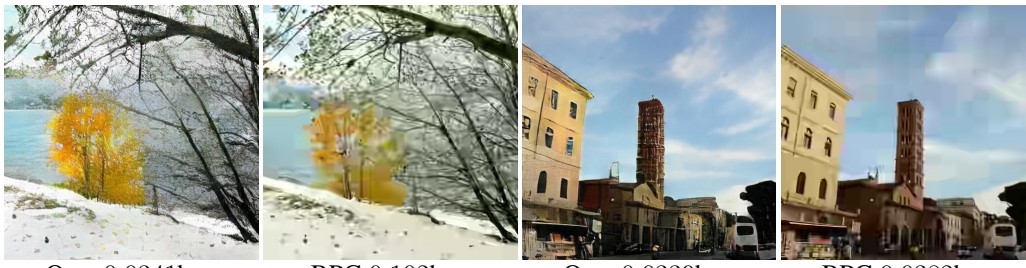

| Ours 0.0341bpp | BPG 0.102bpp | Ours 0.0339bpp | BPG 0.0382bpp |

Figure 3: Visual example of images from RAISE1k produced by our GC network with $C = 4$ along with the corresponding results for BPG.

**User study:** In the extreme compression regime realized by our GC models, where texture and sometimes even more abstract image content is synthesized, common reconstruction quality measures such as PSNR and MS-SSIM arguably lose significance as they penalize changes in local structure rather than assessing preservation of the global image content (this also becomes apparent by comparing reconstructions produced by our GC model with those obtained by the MSE baseline and BPG, see Fig. 2). Indeed, measuring PSNR between synthesized and real texture patches essentially quantifies the variance of the texture rather than the visual quality of the synthesized texture.

To quantitatively evaluate the perceptual quality of our GC models in comparison with BPG and AEDC (for Cityscapes) we therefore conduct a user study using Amazon Mechanical Turk (AMT).[3] We consider two GC models with $C = 4, 8$ trained on Open Images, three GC $(D^+)$ models with $C = 2, 4, 8$ trained on Cityscapes, and BPG at rates ranging from $0.045$ to $0.12$ bpp. Questionnaires are composed by combining the reconstructions produced by the selected GC model for all testing images with the corresponding reconstruction produced by the competing baseline model side-by-side (presenting the reconstructions in random order). The original image is shown along with the reconstructions, and the pairwise comparisons are interleaved with 3 probing comparisons of an additional uncompressed image from the respective testing set with an obviously JPEG-compressed version of that image. 20 randomly selected unique users are asked to indicate their preference for each pair of reconstructions in the questionnaire, resulting in a total of 480 ratings per pairing of methods for Kodak, and 400 ratings for RAISE1K and Cityscapes. For each pairing of methods, we report the mean preference score as well as the standard error (SE) of the per-user mean preference percentages. Only users correctly identifying the original image in all probing comparisons are taken into account for the mean preference percentage computation. To facilitate comparisons for future works, we will release all images used in the user studies.

**Semantic quality of SC models:** The issues with PSNR and MS-SSIM for evaluating the quality of generated content described in the previous paragraph become even more severe for SC models as a large fraction of the image content is generated from a semantic label map. Following image translation works Isola et al. (2017); Wang et al. (2018), we therefore measure the capacity of our SC models to preserve the image semantics in the synthesized regions and plausibly blend them with the preserved regions—the objective SC models are actually trained for. Specifically, we use PSPNet (Zhao et al., 2016) and compute the mean intersection-over-union (IoU) between the label map obtained for the decompressed validation images and the ground truth label map. For reference we also report this metric for baselines that do not use semantic label maps for training and/or deployment.

## 6 RESULTS

### 6.1 GENERATIVE COMPRESSION

Fig. 4 shows the mean preference percentage obtained by our GC models compared to BPG at different rates, on the Kodak and the RAISE1K data set. In addition, we report the mean preference percentage for GC models compared to BPG and AEDC on Cityscapes. Example validation images for side-by-side comparison of our method with BPG for images from the Kodak, RAISE1K, and Cityscapes data set can be found in Figs. 1, 3, and 2, respectively. Furthermore, we perform extensive visual comparisons of all our methods and the baselines, presented in Appendix F.

---

[3]https://www.mturk.com/

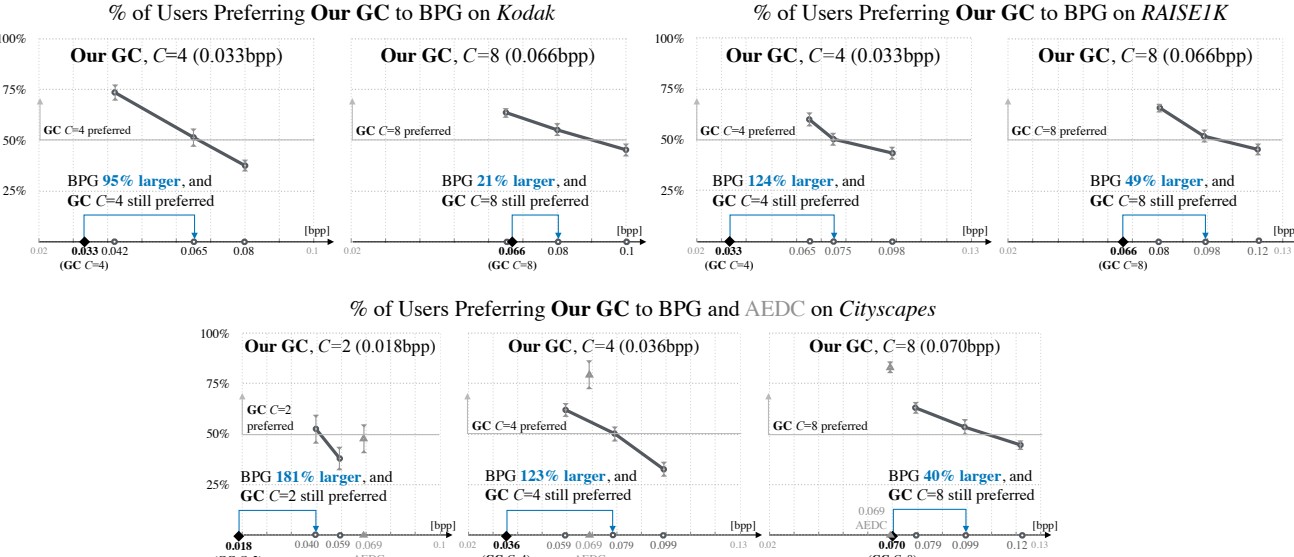

Figure 4: User study results evaluating our GC models on Kodak, RAISE1K (top) and Cityscapes (bottom). For Kodak and RAISE1K, we use GC models trained on Open Images, without any semantic label maps. For Cityscapes, we used GC $(D^+)$, using semantic label maps only for $D$ and only during training. The standard error is computed over per-user mean preference percentages. The blue arrows visualize how many more bits BPG uses when $> 50\%$ users still prefer our result.

Our GC models with $C = 4$ are preferred to BPG even when images produced by BPG use 95% and 124% more bits than those produced by our models for Kodak and RAISE1K, respectively. Notably this is achieved even though there is a distribution shift between the training and testing set (recall that these GC models are trained on Open Images). The gains of domain-specificity and semantic label maps (for training) becomes apparent from the results on Cityscapes: Our GC models with $C = 2$ are preferred to BPG even when the latter uses 181% more bits. For $C = 4$ the gains on Cityscapes are comparable to those obtained for GC on RAISE1K. For all three data sets, BPG requires between 21 and 49% more bits than our GC models with $C = 8$.

**Discussion:** The GC models produce images with much finer detail than BPG, which suffers from smoothed patches and blocking artifacts. In particular, the GC models convincingly reconstruct texture in natural objects such as trees, water, and sky, and is most challenged with scenes involving humans. AEDC and the MSE baseline both produce blurry images.

We see that the gains of our models are maximal at extreme bitrates, with BPG needing 95–181% more bits for the $C = 2, 4$ models on the three datasets. For $C = 8$ gains are smaller but still very large (BPG needing 21–49% more bits). This is expected, since as the bitrate increases the classical compression measures (PSNR/MS-SSIM) become more meaningful—and our system does not employ the full complexity of current state-of-the-art systems, as discussed next.

**State-of-the-art on Kodak:** We give an overview of relevant recent learned compression methods and their differences to our GC method and BPG in Table 1 in the Appendix. Rippel & Bourdev (2017) also used GANs (albeit a different formulation) and were state-of-the-art in MS-SSIM in 2017, while the concurrent work of Minnen et al. (2018) is the current state-of-the-art in image compression in terms of classical metrics (PSNR and MS-SSIM) when measured on the Kodak dataset (Kodak). Notably, all methods except ours (BPG, Rippel et al., and Minnen et al.) employ adaptive arithmetic coding using context models for improved compression performance. Such models could also be implemented for our system, and have led to additional savings of 10% in Mentzer et al. (2018). Since Rippel et al. and Minnen et al. have only released a selection of their decoded images (for 3 and 4, respectively, out of the 24 Kodak images), and at significantly higher bitrates, a comparison with a user study is not meaningful. Instead, we try to qualitatively put our results into context with theirs.

In Figs. 12–14 in the Appendix, we compare qualitatively to Rippel & Bourdev (2017). We can observe that even though Rippel & Bourdev (2017) use 29–179% more bits, our models produce images of comparable or better quality.

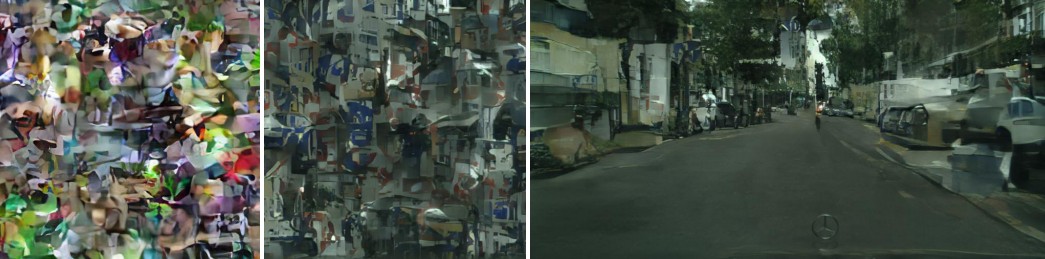

| Uniform, Open Images | Uniform, Cityscapes | Code samples generated by WGAN-GP, Cityscapes |

Figure 5: Sampling codes $\hat{w}$ uniformly (left), and generating them with a WGAN-GP (right).

In Figs. 15–18, we show a qualitative comparison of our results to the images provided by the concurrent work of Minnen et al. (2018), as well as to BPG (Bellard) on those images. First, we see that BPG is still visually competitive with the current state-of-the-art, which is consistent with moderate $8.41\%$ bitrate savings being reported by Minnen et al. (2018) in terms of PSNR. Second, even though we use much fewer bits compared to the example images available from Minnen et al. (2018), for some of them (Figs. 15 and 16) our method can still produce images of comparable visual quality.

Given the dramatic bitrate savings we achieve according to the user study (BPG needing 21–181% more bits), and the competitiveness of BPG to the most recent state-of-the-art (Minnen et al., 2018), we conclude that our proposed system presents a **significant step forward** for visually pleasing compression at extreme bitrates.

**Sampling the compressed representations:** In Fig. 5 we explore the representation learned by our GC models (with $C = 4$), by sampling the (discrete) latent space of $\hat{w}$. When we sample uniformly, and decode with our GC model into images, we obtain a "soup of image patches" which reflects the domain the models were trained on (e.g. street sign and building patches on Cityscapes). Note that we should not expect these outputs to look like normal images, since nothing forces the encoder output $\hat{w}$ to be uniformly distributed over the discrete latent space.

However, given the low dimensionality of $\hat{w}$ ($32 \times 64 \times 4$ for $512 \times 1024$px Cityscape images), it would be interesting to try to learn the true distribution. To this end, we perform a simple experiment and train an improved Wasserstein GAN (WGAN-GP) (Gulrajani et al., 2017) on $\hat{w}$ extracted from Cityscapes, using default parameters and a ResNet architecture.[4] By feeding our GC model with samples from the WGAN-GP generator, we easily obtain a powerful generative model, which generates sharp $1024 \times 512$px images *from scratch*. We think this could be a promising direction for building high-resolution generative models. In Figs. 19–21 in the Appendix, we show more samples, and samples obtained by feeding the MSE baseline with uniform and learned code samples. The latter yields noisier "patch soups" and much blurrier image samples than our GC network.

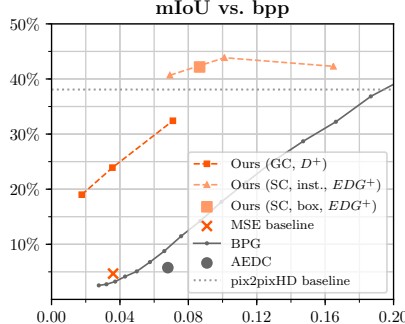

Figure 6: Mean IoU as a function of bpp on the Cityscapes validation set for our GC and SC networks, and for the MSE baseline. We show both SC modes: RI (inst.), RB (box). $D^+$ annotates models where instance semantic label maps are fed to the discriminator (only during training); $EDG^+$ indicates that semantic label maps are used both for training and deployment. The pix2pixHD baseline (Wang et al., 2018) was trained from scratch for 50 epochs, using the same downsampled $1024 \times 512$px training images as for our method.

## 6.2 SELECTIVE GENERATIVE COMPRESSION

Fig. 6 shows the mean IoU on the Cityscapes validation set as a function of bpp for SC networks with $C = 2, 4, 8$, along with the values obtained for the baselines. Additionally, we plot mean IoU for GC with semantic label maps fed to the discriminator ($D^+$), and the MSE baseline.

---

[4]We only adjusted the architecture to output $32 \times 64 \times 4$ tensors instead of $64 \times 64 \times 3$ RGB images.

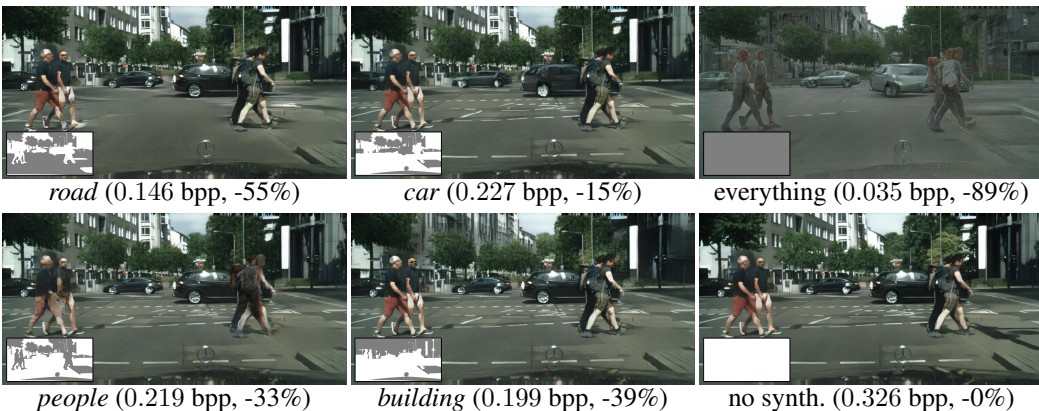

*road* (0.146 bpp, -55%)  *car* (0.227 bpp, -15%)  everything (0.035 bpp, -89%)

*people* (0.219 bpp, -33%)  *building* (0.199 bpp, -39%)  no synth. (0.326 bpp, -0%)

Figure 7: Synthesizing different classes using our SC network with $C = 8$. In each image except for *no synthesis*, we additionally synthesize the classes *vegetation, sky, sidewalk, ego vehicle, wall*. The heatmaps in the lower left corners show the synthesized parts in gray. We show the bpp of each image as well as the relative savings due to the selective generation.

In Fig. 7 we present example Cityscapes validation images produced by the SC network trained in the RI mode with $C = 8$, where different semantic classes are preserved. More visual results for the SC networks trained on Cityscapes can be found in Appendix F.7, including results obtained for the RB operation mode and by using semantic label maps estimated from the input image via PSPNet (Zhao et al., 2017).

**Discussion:** The quantitative evaluation of the semantic preservation capacity (Fig. 6) reveals that the SC networks preserve the semantics somewhat better than pix2pixHD, indicating that the SC networks faithfully generate texture from the label maps and plausibly combine generated with preserved image content. The mIoU of BPG, AEDC, and the MSE baseline is considerably lower than that obtained by our SC and GC models, which can arguably be attributed to blurring and blocking artifacts. However, it is not surprising as these baseline methods do not use label maps during training and prediction.

In the SC operation mode, our networks manage to seamlessly merge preserved and generated image content both when preserving object instances and boxes crossing object boundaries (see Appendix F.7). Further, our networks lead to reductions in bpp of 50% and more compared to the same networks without synthesis, while leaving the visual quality essentially unimpaired, when objects with repetitive structure are synthesized (such as trees, streets, and sky). In some cases, the visual quality is even better than that of BPG at the same bitrate. The visual quality of more complex synthesized objects (e.g. buildings, people) is worse. However, this is a limitation of current GAN technology rather than our approach. As the visual quality of GANs improves further, SC networks will as well. Notably, the SC networks can generate entire images from the semantic label map only.

Finally, the semantic label map, which requires 0.036 bpp on average for the downscaled $1024 \times 512$px Cityscapes images, represents a relatively large overhead compared to the storage cost of the preserved image parts. This cost vanishes as the image size increases, since the semantic mask can be stored as an image dimension-independent vector graphic.

## 7 CONCLUSION

We proposed and evaluated a GAN-based framework for learned compression that significantly outperforms prior works for low bitrates in terms of visual quality, for compression of natural images. Furthermore, we demonstrated that constraining the application domain to street scene images leads to additional storage savings, and we explored combining synthesized with preserved image content with the potential to achieve even larger savings. Interesting directions for future work are to develop a mechanism for controlling spatial allocation of bits for GC (e.g. to achieve better preservation of faces; possibly using semantic label maps), and to combine SC with saliency information to determine what regions to preserve. In addition, the sampling experiments presented in Sec. 6.1 indicate that combining our GC compression approach with GANs to (unconditionally) generate compressed representations is a promising avenue to learn high-resolution generative models.

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

## A    COMPARISON WITH STATE-OF-THE-ART

|  | BPG | Rippel et al. (2017) | Minnen et al. (2018) | Ours (GC) |
|---|---|---|---|---|
| Learned | No | Yes | Yes | Yes |
| Arithmetic encoding | Adaptive | Adaptive | Adaptive | Static |
| Context model | CABAC | Autoregressive | Autoregressive | None |
| Visualized bitrates [bpp][5] | all[6] | 0.08– | 0.12– | 0.033–0.066 |
| GAN | No | Non-standard | No | f-div. based |
| S.o.t.a. in MS-SSIM | No | No | Yes | No |
| S.o.t.a. in PSNR | No | No | Yes | No |
| Savings to BPG in PSNR |  |  | 8.41% |  |
| Savings to BPG in User Study |  |  |  | 17.2–48.7% |

Table 1: Overview of differences between (Minnen et al., 2018) (s.o.t.a. in MS-SSIM and PSNR), to BPG (previous s.o.t.a. in PSNR) and (Rippel & Bourdev, 2017) (s.o.t.a. in MS-SSIM in 2017, also used GANs).

## B    COMPRESSION DETAILS

When encoding the channels of $\hat{w}$ to a bit-stream, we use an arithmetic encoder where frequencies are stored for each channel separately and then encode them in a static (non-adaptive) manner, similar to Agustsson et al. (2017). In our experiments, this leads to $8.8\%$ smaller bitrates compared to the upper bound.

We compress the semantic label map for SC by quantizing the coordinates in the vector graphic to the image grid and encoding coordinates relative to preceding coordinates when traversing object boundaries (rather than relative to the image frame). The so-obtained bitstream is then compressed using arithmetic coding.

To ensure fair comparison, we do not count header sizes for any of the baseline methods throughout.

## C    ARCHITECTURE DETAILS

For the GC, the encoder $E$ convolutionally processes the image $x$ and optionally the label map $s$, with spatial dimension $W \times H$, into a feature map of size $W/16 \times H/16 \times 960$ (with 6 layers, of which four have 2-strided convolutions), which is then projected down to $C$ channels (where $C \in \{2, 4, 8\}$ is much smaller than 960). This results in a feature map $w$ of dimension $W/16 \times H/16 \times C$, which is quantized over $L$ centers to obtain the discrete $\hat{w}$. The generator $G$ projects $\hat{w}$ up to 960 channels, processes these with 9 residual units (He et al., 2016) at dimension $W/16 \times H/16 \times 960$, and then mirrors $E$ by convolutionally processing the features back to spatial dimensions $W \times H$ (with transposed convolutions instead of strided ones).

Similar to $E$, the feature extractor $F$ for SC processes the semantic map $s$ down to the spatial dimension of $\hat{w}$, which is then concatenated to $\hat{w}$ for generation. In this case, we consider slightly higher bitrates and downscale by $8\times$ instead of $16\times$ in the encoder $E$, such that $\dim(\hat{w}) = W/8 \times H/8 \times C$. The generator then first processes $\hat{w}$ down to $W/16 \times H/16 \times 960$ and then proceeds as for GC.

For both GC and SC, we use the multi-scale architecture of (Wang et al., 2018) for the discriminator $D$, which measures the divergence between $p_x$ and $p_{G(z)}$ both locally and globally.

We adopt the notation from (Wang et al., 2018) to describe our encoder and generator/decoder architectures and additionally use q to denote the quantization layer (see Sec. 3 for details). The output of q is encoded and stored.

- **Encoder GC:** `c7s1-60,d120,d240,d480,d960,c3s1-`$C$`,q`

---

[5]This refers to the bitrates of decoded images the authors have made available.

[6]Code available, image can be compressed from extreme bpps ($< 0.1$bpp) to lossless.

- **Encoders SC:**

  - Semantic label map encoder: `c7s1-60,d120,d240,d480,d960`

  - Image encoder: `c7s1-60,d120,d240,d480,c3s1-`$C$`,q,c3s1-480,d960`

  The outputs of the semantic label map encoder and the image encoder are concatenated and fed to the generator/decoder.

- **Generator/decoder:** `c3s1-960,R960,R960,R960,R960,R960,R960,R960,`
  `R960,R960,u480,u240,u120,u60,c7s1-3`

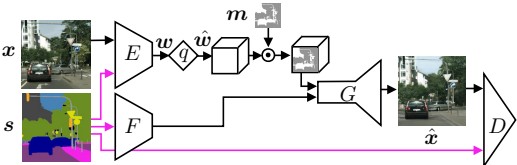

Figure 8: Structure of the proposed SC network. $E$ is the encoder for the image $x$ and the semantic label map $s$. $q$ quantizes the latent code $w$ to $\hat{w}$. The subsampled heatmap multiplies $\hat{w}$ (pointwise) for spatial bit allocation. $G$ is the generator/decoder, producing the decompressed image $\hat{x}$, and $D$ is the discriminator used for adversarial training. $F$ extracts features from $s$ .

## D  TRAINING DETAILS

We employ the ADAM optimizer (Kingma & Ba, 2014) with a learning rate of $0.0002$ and set the mini-batch size to 1. Our networks are trained for 150000 iterations on Cityscapes and for 280000 iterations on Open Images. For normalization we used instance normalization (Ulyanov et al., 2016), except in the second half of the Open Images training, we train the generator/decoder with fixed batch statistics (as implemented in the test mode of batch normalization (Ioffe & Szegedy, 2015)), since we found this reduced artifacts and color shift.

## E  DATASET AND PREPROCESSING DETAILS

To train GC models (which do not require semantic label maps, neither during training nor for deployment) for compression of diverse natural images, we use 200k images sampled randomly from the *Open Images* data set (Krasin et al., 2017) (9M images). The training images are rescaled so that the longer side has length 768px, and images for which rescaling does not result in at least $1.25\times$ downscaling as well as high saturation images (average S $> 0.9$ or V $> 0.8$ in HSV color space) are discarded (resulting in an effective training set size of 188k). We evaluate these models on the Kodak image compression dataset (Kodak) (24 images, $768 \times 512$px), which has a long tradition in the image compression literature and is still the most frequently used dataset for comparisons of learned image compression methods. Additionally, we evaluate our GC models on 20 randomly selected images from the *RAISE1K* data set (Dang-Nguyen et al., 2015), a real-world image dataset consisting of 8156 high-resolution RAW images (we rescale the images such that the longer side has length 768px). To investigate the benefits of having a somewhat constrained application domain and semantic labels at training time, we also train GC models with semantic label maps on the *Cityscapes* data set (Cordts et al., 2016) (2975 training and 500 validation images, 34 classes, $2048 \times 1024$px resolution) consisting of street scene images and evaluate it on 20 randomly selected validation images (without semantic labels). Both training and validation images are rescaled to $1024 \times 512$px resolution.

To evaluate the proposed SC method (which requires semantic label maps for training and deployment) we again rely on the Cityscapes data set. Cityscapes was previously used to generate images form semantic label maps using GANs (Isola et al., 2017; Zhu et al., 2017). The preprocessing for SC is the same as for GC.

## F VISUALS

In the following Sections, F.1, F.2, F.3, we show the first five images of each of the three datasets we used for the user study, next to the outputs of BPG at similar bitrates.

Secs. F.4 and F.5 provide visual comparisons of our GC models with Rippel & Bourdev (2017) and Minnen et al. (2018), respectively, on a subset of images form the Kodak data set.

In Section F.6, we show visualizations of the latent representation of our GC models.

Finally, Section F.7 presents additional visual results for SC.

## F.1 GENERATIVE COMPRESSION ON KODAK

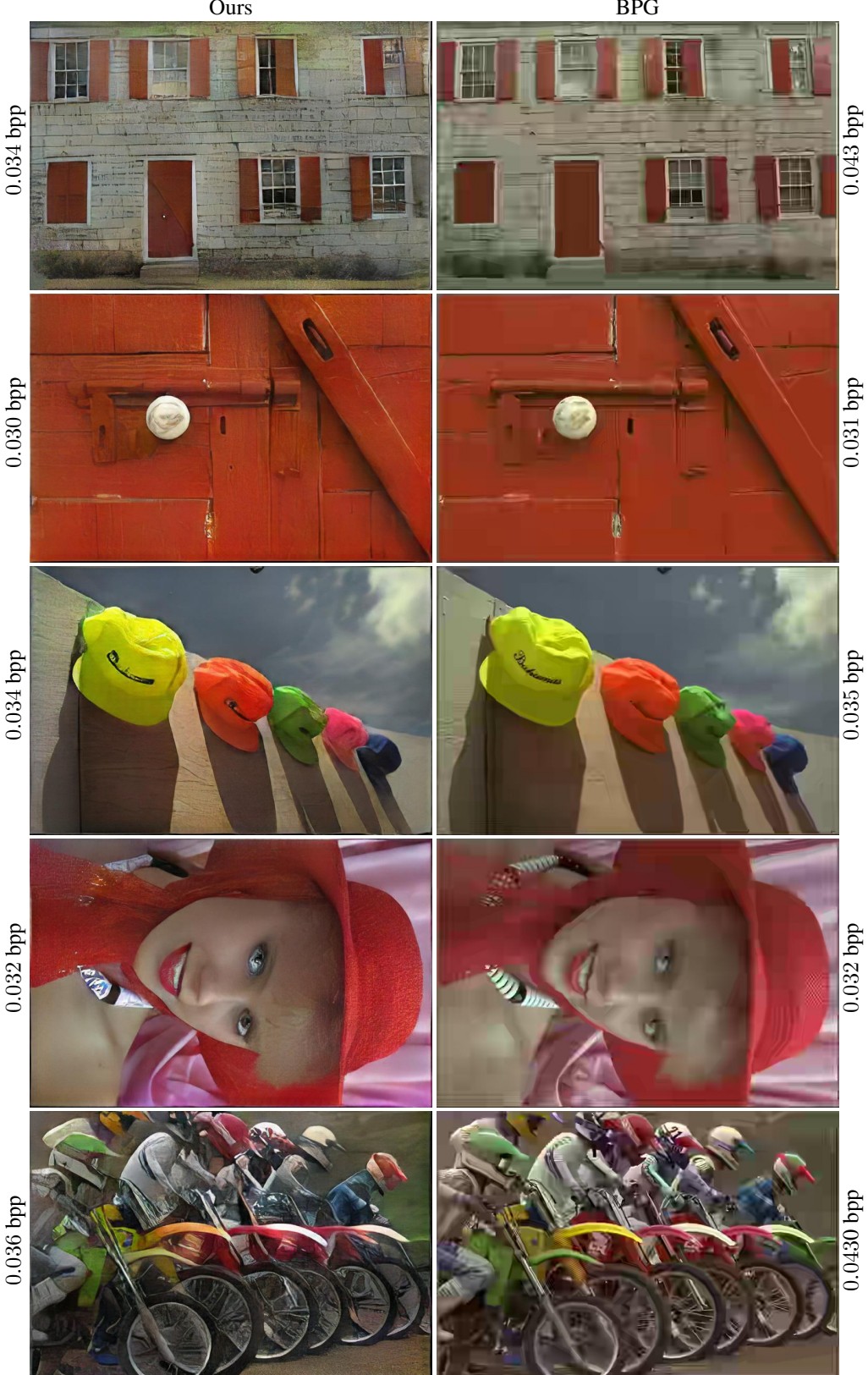

Figure 9: First 5 images of the Kodak data set, produced by our GC model with $C = 4$ and BPG.

F.2    GENERATIVE COMPRESSION ON RAISE1K

Ours                                                                                          BPG

Figure 10: First 5 images of RAISE1k, produced by our GC model with $C = 4$ and BPG.

## F.3 GENERATIVE COMPRESSION ON CITYSCAPES

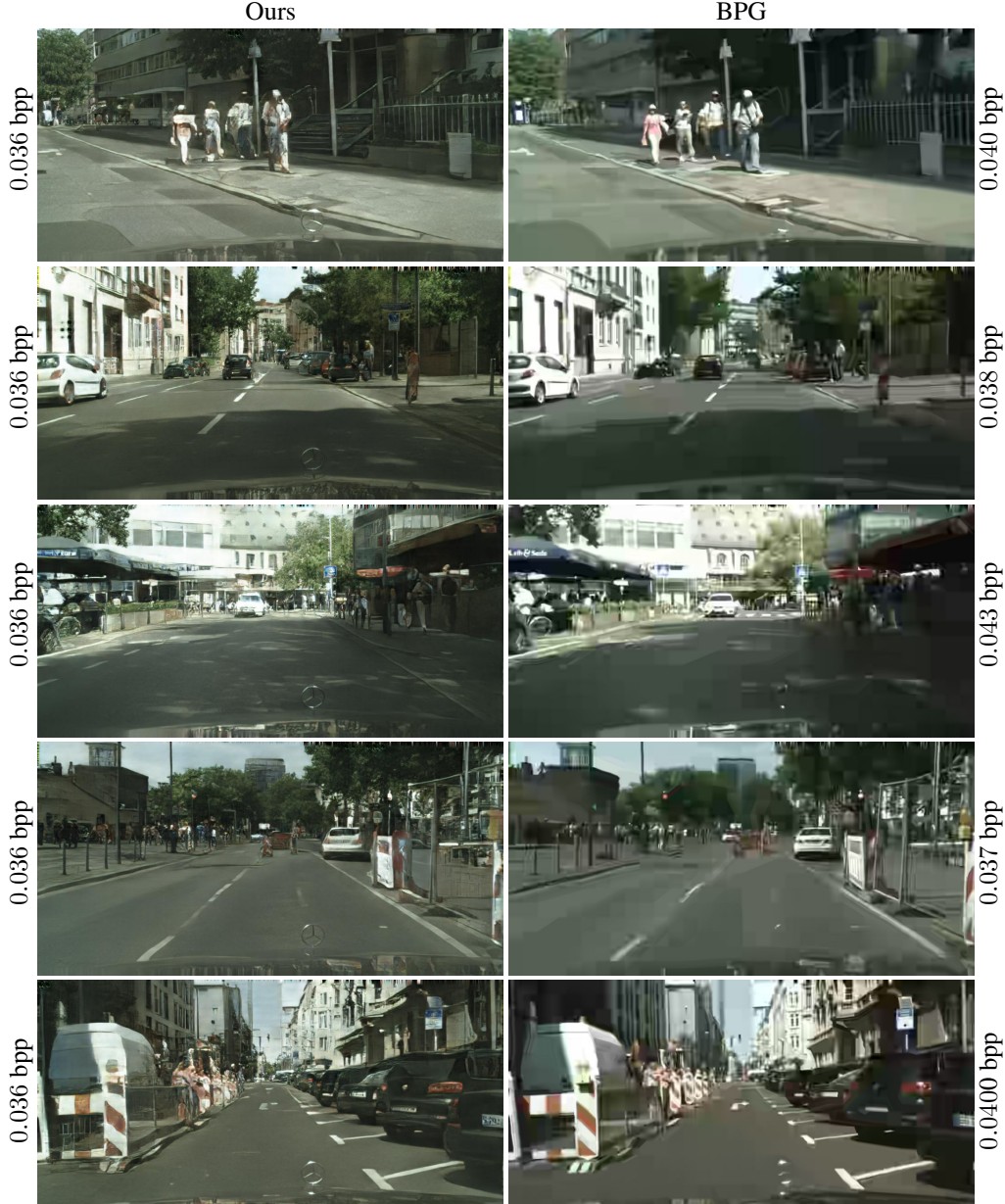

Figure 11: First 5 images of Cityscapes, produced by our GC model with $C = 4$ and BPG.

### F.4 COMPARISON TO RIPPEL & BOURDEV (2017)

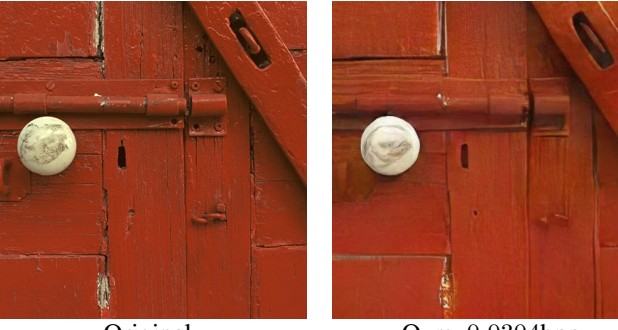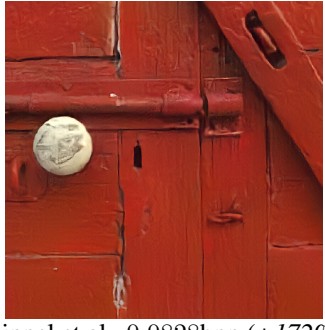

| Original | Ours, 0.0304bpp | Rippel et al., 0.0828bpp (*+172%*) |

Figure 12: Our model loses more texture but has less artifacts on the knob. Overall, it looks comparable to the output of Rippel & Bourdev (2017), using significantly fewer bits.

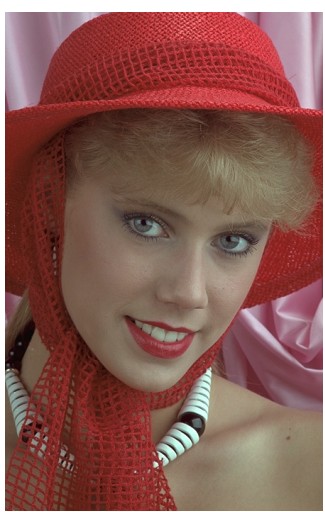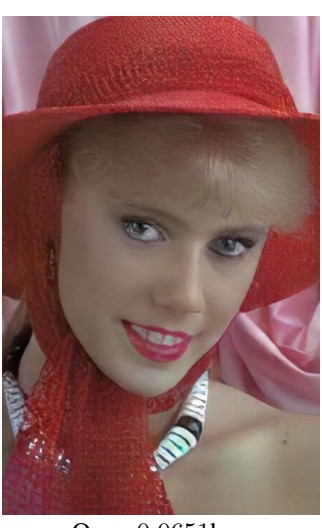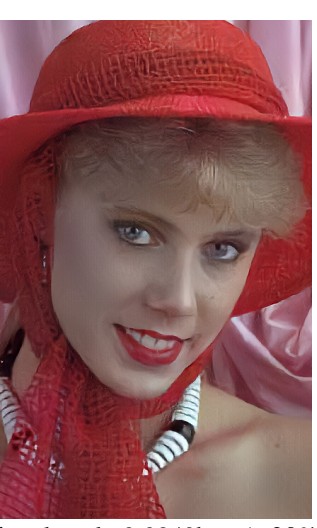

| Original | Ours, 0.0651bpp | Rippel et al., 0.0840bpp (*+29%*) |

Figure 13: Notice that compared to Rippel & Bourdev (2017), our model produces smoother lines at the jaw and a smoother hat, but proides a worse reconstruction of the eye.

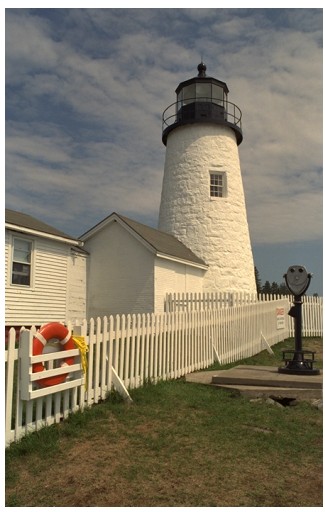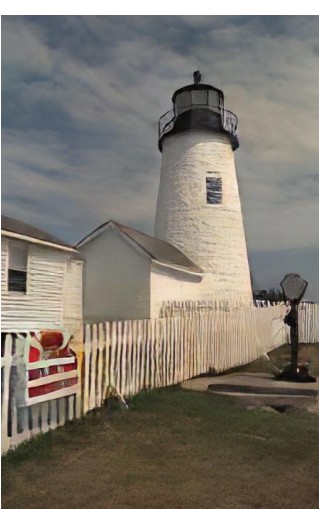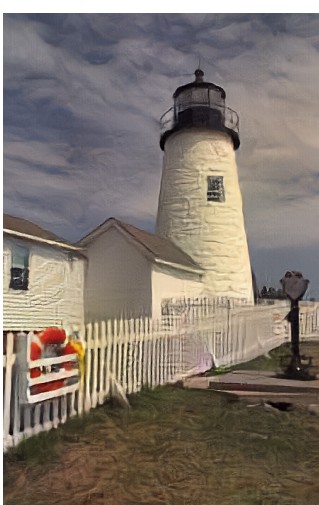

| Original | Ours, 0.0668bpp | Rippel et al., 0.0928bpp (*+39%*) |

Figure 14: Notice that our model produces much better sky and grass textures than Rippel & Bourdev (2017), and also preserves the texture of the light tower more faithfully.

## F.5 Comparison to Minnen et al. (2018)

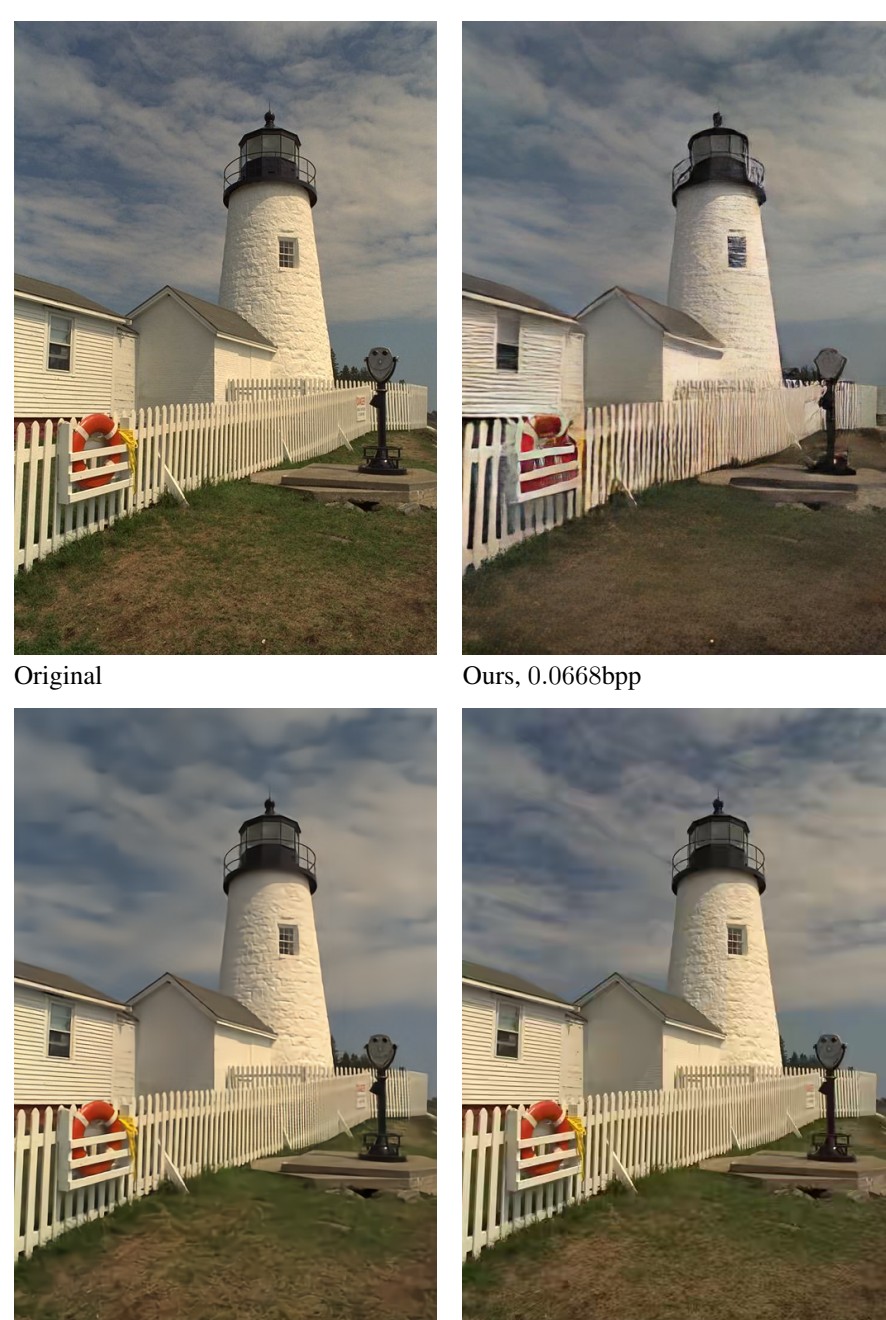

Original                                    Ours, 0.0668bpp

Minnen et al., 0.221bpp *230% larger*       BPG, 0.227bpp

Figure 15: Notice that our model yields sharper grass and sky, but a worse reconstruction of the fence and the lighthouse compared to Minnen et al. (2018). Compared to BPG, Minnen et al. produces blurrier grass, sky and lighthouse but BPG suffers from ringing artifacts on the roof of the second building and the top of the lighthouse.

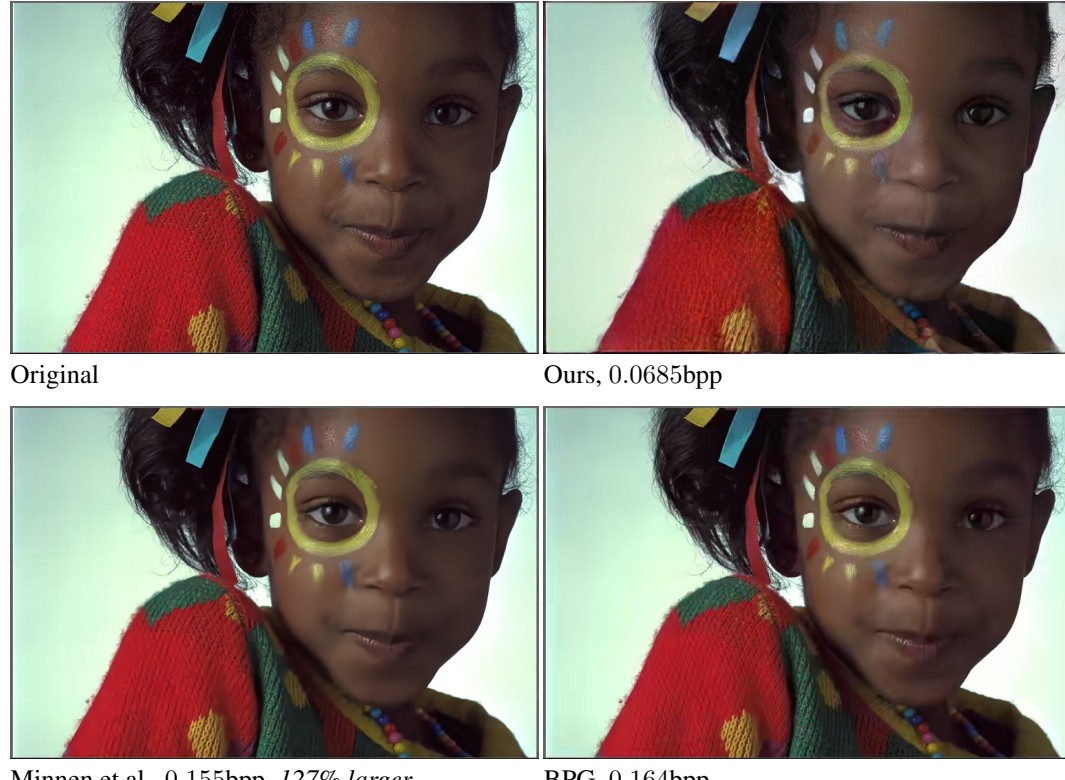

Original                                    Ours, 0.0685bpp

Minnen et al., 0.155bpp, *127% larger*      BPG, 0.164bpp

Figure 16: Our model produces an overall sharper face compared to Minnen et al. (2018), but the texture on the cloth deviates more from the original. Compared to BPG, Minnen et al. has a less blurry face and fewer artifacts on the cheek.

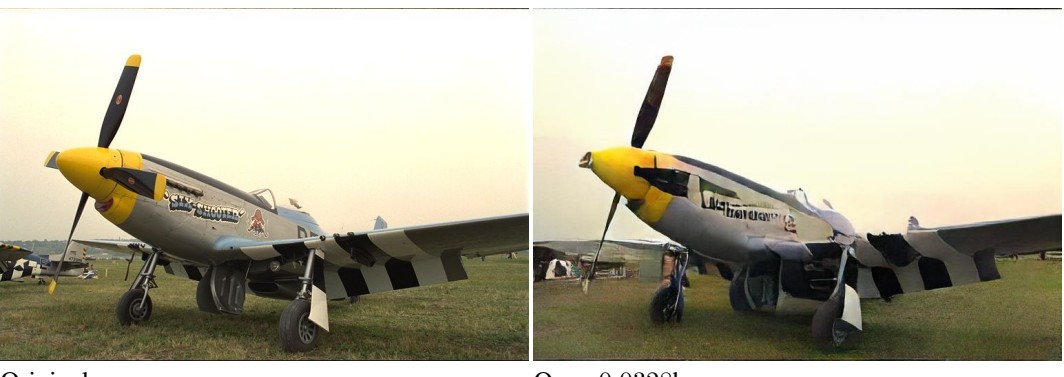

Original                                    Ours, 0.0328bpp

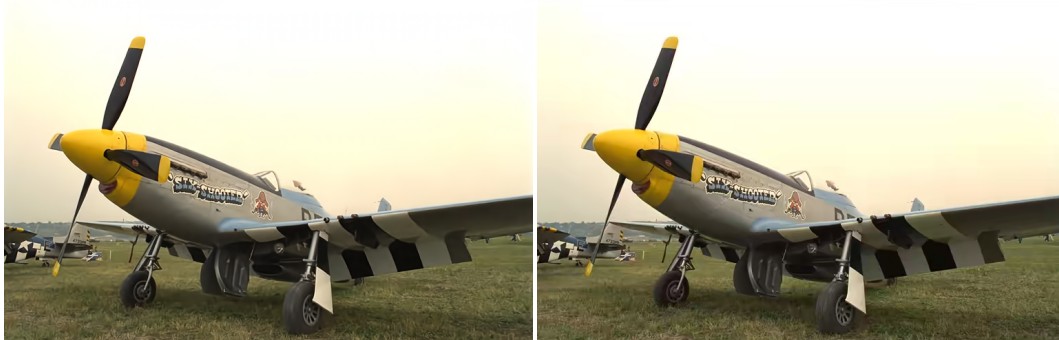

Minnen et al., 0.246bpp, *651% larger*      BPG, 0.248bpp

Figure 17: Here we obtain a significantly worse reconstruction than Minnen et al. (2018) and BPG, but use only a fraction of the bits. Between BPG and Minnen et al., it is hard to see any differences.

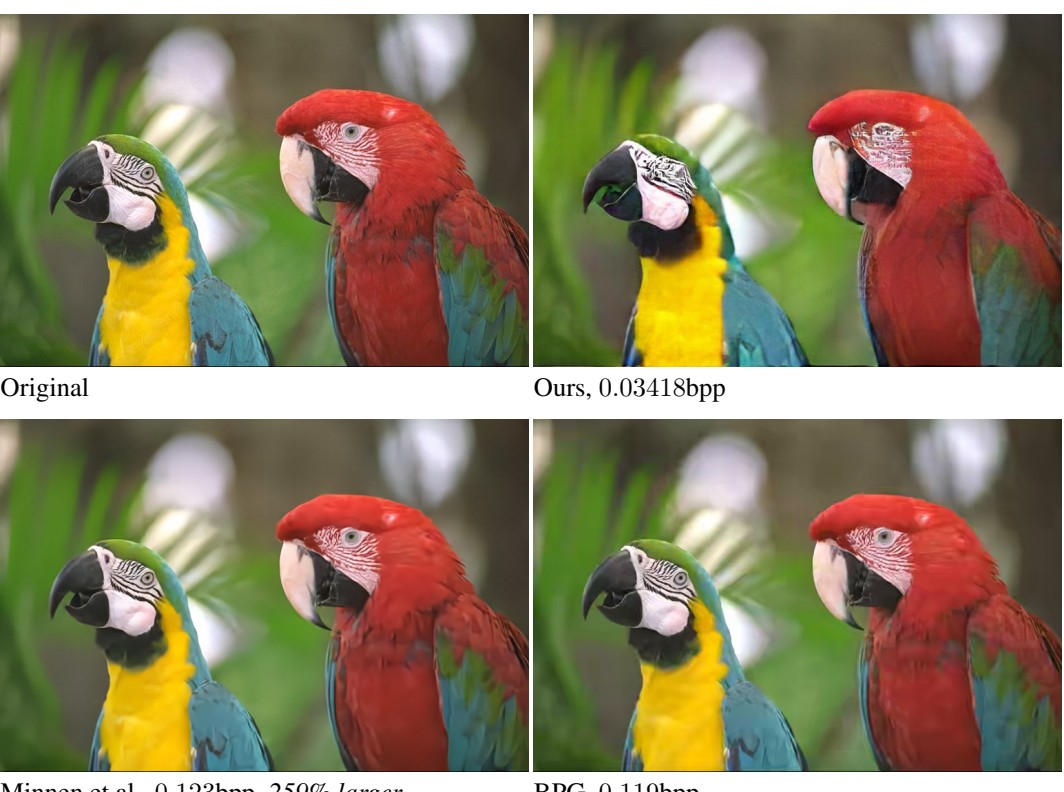

Original                Ours, 0.03418bpp

Minnen et al., 0.123bpp, *259% larger*,   BPG, 0.119bpp

Figure 18: Here we obtain a significantly worse reconstruction compared to Minnen et al. (2018) and BPG, but use only a fraction of the bits. Compared to BPG, Minnen et al.has a smoother background but less texture on the birds.

F.6    SAMPLING THE COMPRESSED REPRESENTATIONS

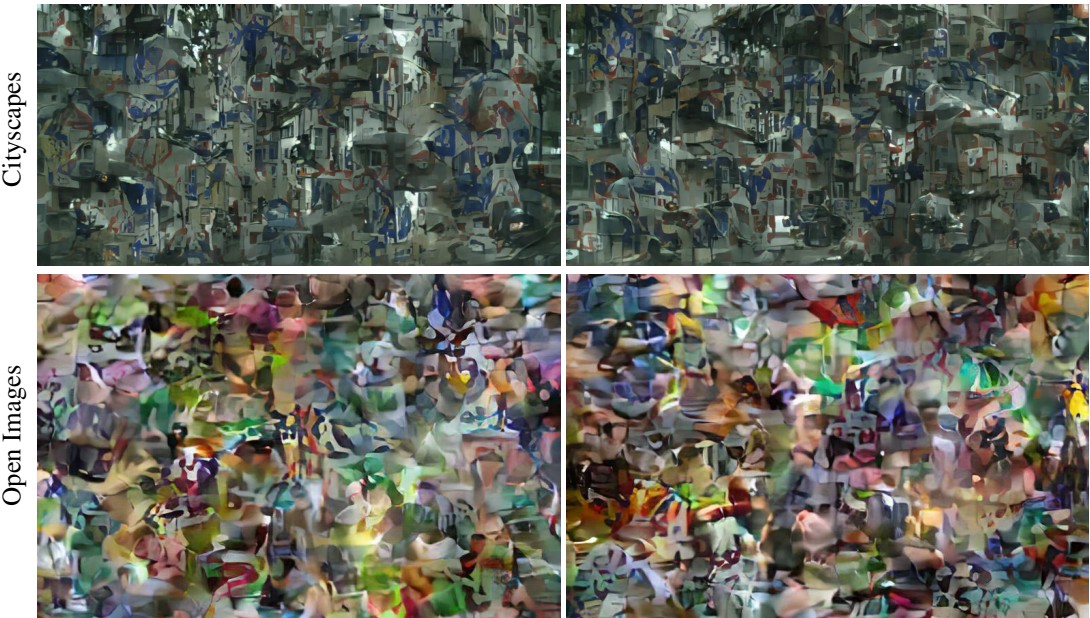

Figure 19: We uniformly sample codes from the (discrete) latent space $\hat{w}$ of our generative compression models (GC with $C = 4$) trained on Cityscapes and Open Images. The Cityscapes model outputs domain specific patches (street signs, buildings, trees, road), whereas the Open Images samples are more colorful and consist of more generic visual patches.

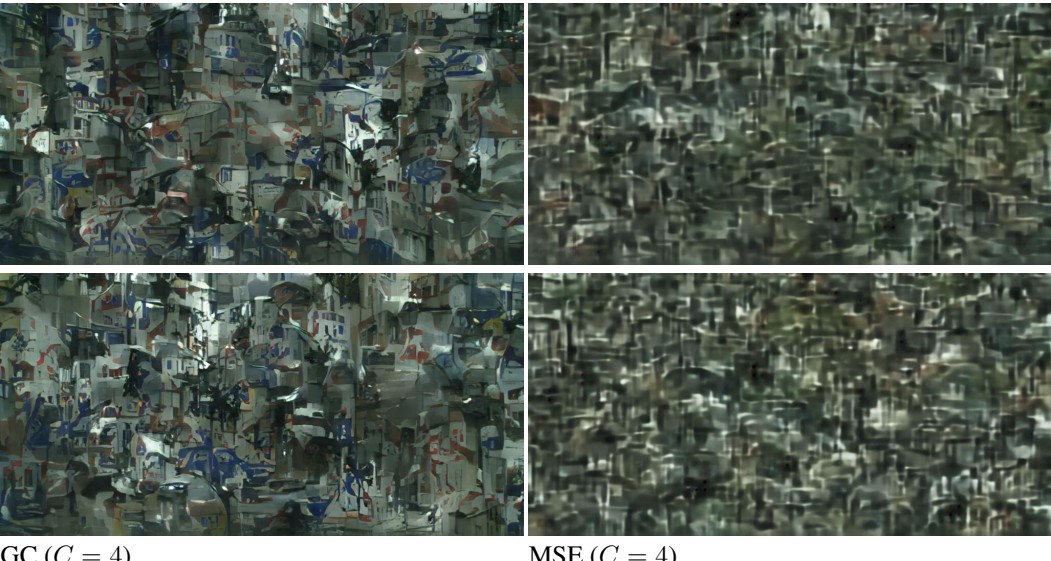

GC ($C = 4$)                                    MSE ($C = 4$)

Figure 20: We train the same architecture with $C = 4$ for MSE and for generative compression on Cityscapes. When uniformly sampling the (discrete) latent space $\hat{w}$ of the models, we see stark differences between the decoded images $G(\hat{w})$. The GC model produces patches that resemble parts of Cityscapes images (street signs, buildings, etc.), whereas the MSE model outputs looks like low-frequency noise.

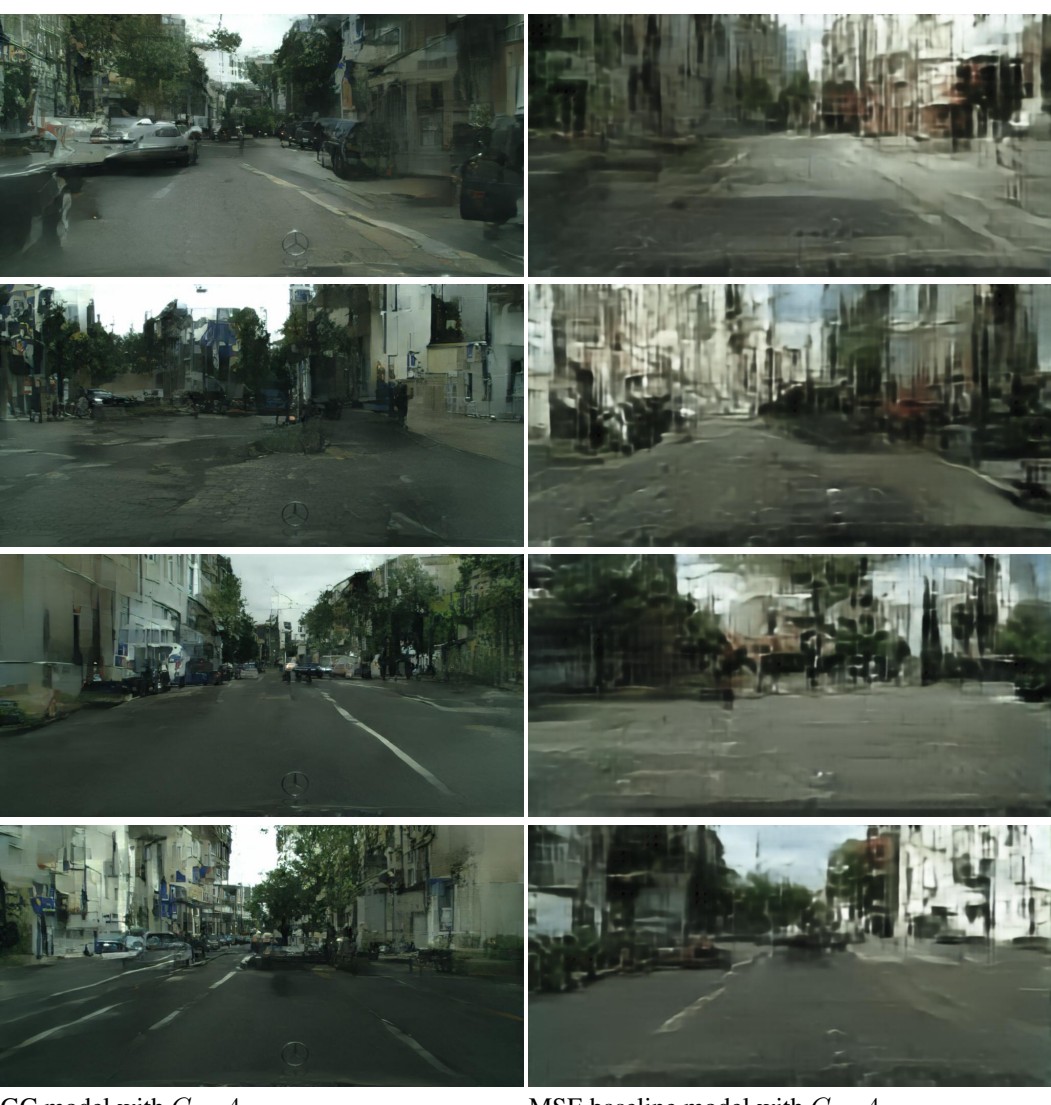

GC model with $C = 4$                                       MSE baseline model with $C = 4$

Figure 21: We experiment with learning the distribution of $\hat{\boldsymbol{w}} = E(\boldsymbol{x})$ by training an improved Wasserstein GAN (Gulrajani et al., 2017). When sampling form the decoder/generator $G$ of our model by feeding it with samples from the improved WGAN generator, we obtain much sharper images than when we do the same with an MSE model.

## F.7 SELECTIVE COMPRESSION ON CITYSCAPES

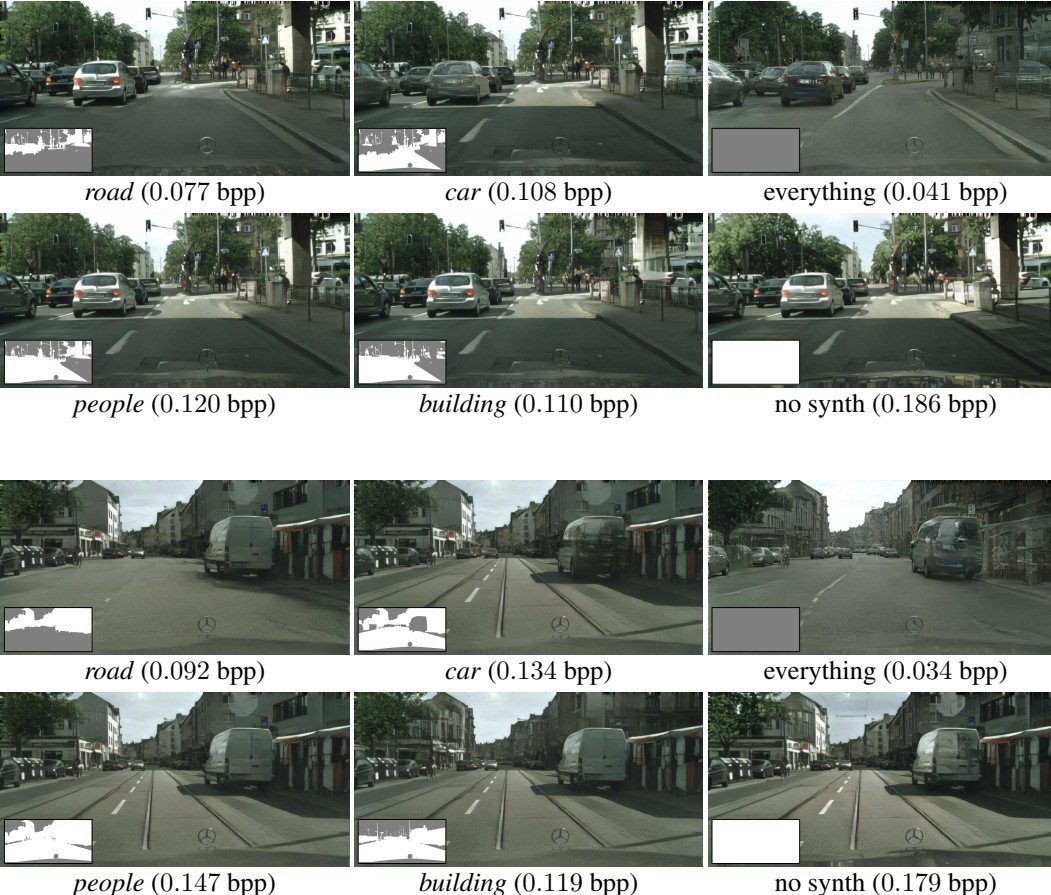

Figure 22: Synthesizing different classes for two different images from Cityscapes, using our SC network with $C = 4$. In each image except for *no synthesis*, we additionally synthesize the classes *vegetation, sky, sidewalk, ego vehicle, wall*.

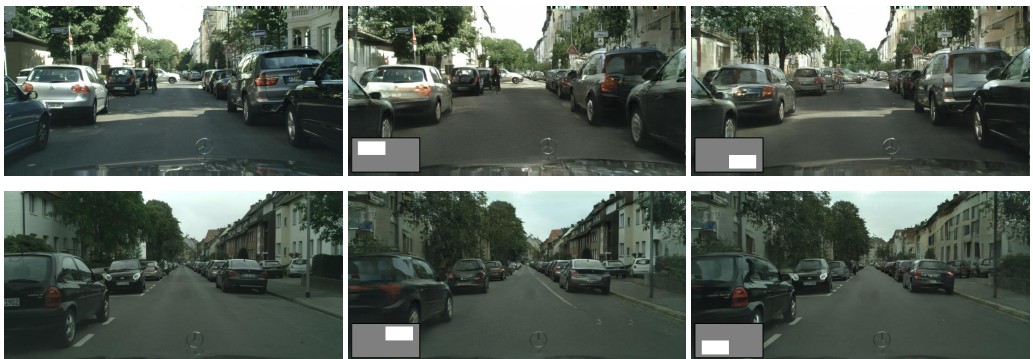

Figure 23: Example images obtained by our SC network ($C = 8$) preserving a box and synthesizing the rest of the image, on Cityscapes. The SC network seamlessly merges preserved and generated image content even in places where the box crosses object boundaries.

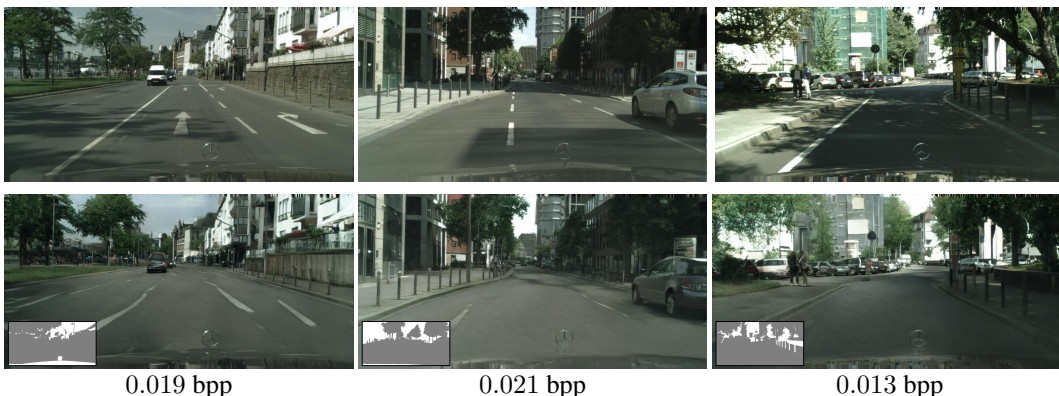

| 0.019 bpp | 0.021 bpp | 0.013 bpp |

Figure 24: Reconstructions obtained by our SC network using semantic label maps estimated from the input image via PSPNet (Zhao et al., 2017).

### F.8 Additional results

We collect here additional results for the discussion with the reviewers, so that they are easily found. We will integrate these results into the paper.

In Table 2 we compute the PSNR on the Cityscapes test set, when varying the entropy constraint (i.e. changing $C$), and the two extremes (a) when MSE is only optimized and (b) when the GAN loss is only optimized. The first three rows shows as the entropy constraint is relaxed, the network can more easily optimize the distortion term leading to a higher PSNR. The fourth row shows that when optimizing for MSE only (see Fig.7 for a qualitative example) we obtain superior PSNR (but at the expense of visual quality with blurry images). The last rows shows that when turning off distortion losses ($\lambda = 0$), the network does optimize reconstruction at all. Here we observe that the GAN "collapses" and outputs repetitive textures (see Fig. 25), suggesting the distortion losses are crucial for stability of training.

In Figures 26&27 we show the loss curves when training our GC, $C = 8$ model on OpenImages(Krasin et al., 2017). We note that the loss fluctuates heavily across iterations due to the small batch size (one), but the smoothed losses are stable. For all our experiments, both on Cityscapes and OpenImages, we kept the weights of the losses and ratio between discriminator/generator iterations constant and at point did our (GC and SC) models collapse during training for either dataset.

| Setting | PSNR (dB) |
|---|---|
| Our GC, $C = 2, H(\hat{\boldsymbol{w}}) < 0.018 bpp$ | 21.46 |
| Our GC, $C = 4, H(\hat{\boldsymbol{w}}) < 0.036$ | 23.17 |
| Our GC, $C = 8, H(\hat{\boldsymbol{w}}) < 0.072$ | 24.93 |
| MSE bl., $C = 4, H(\hat{\boldsymbol{w}}) < 0.036$ | 25.91 |
| GC, $\lambda = 0, C = 8, H(\hat{\boldsymbol{w}}) < 0.072$ | 11.65 |

Table 2: We consider the effect of the GAN loss, the distortion losses and the entropy constraint on the PSNR of the trained model on the Cityscapes dataset.

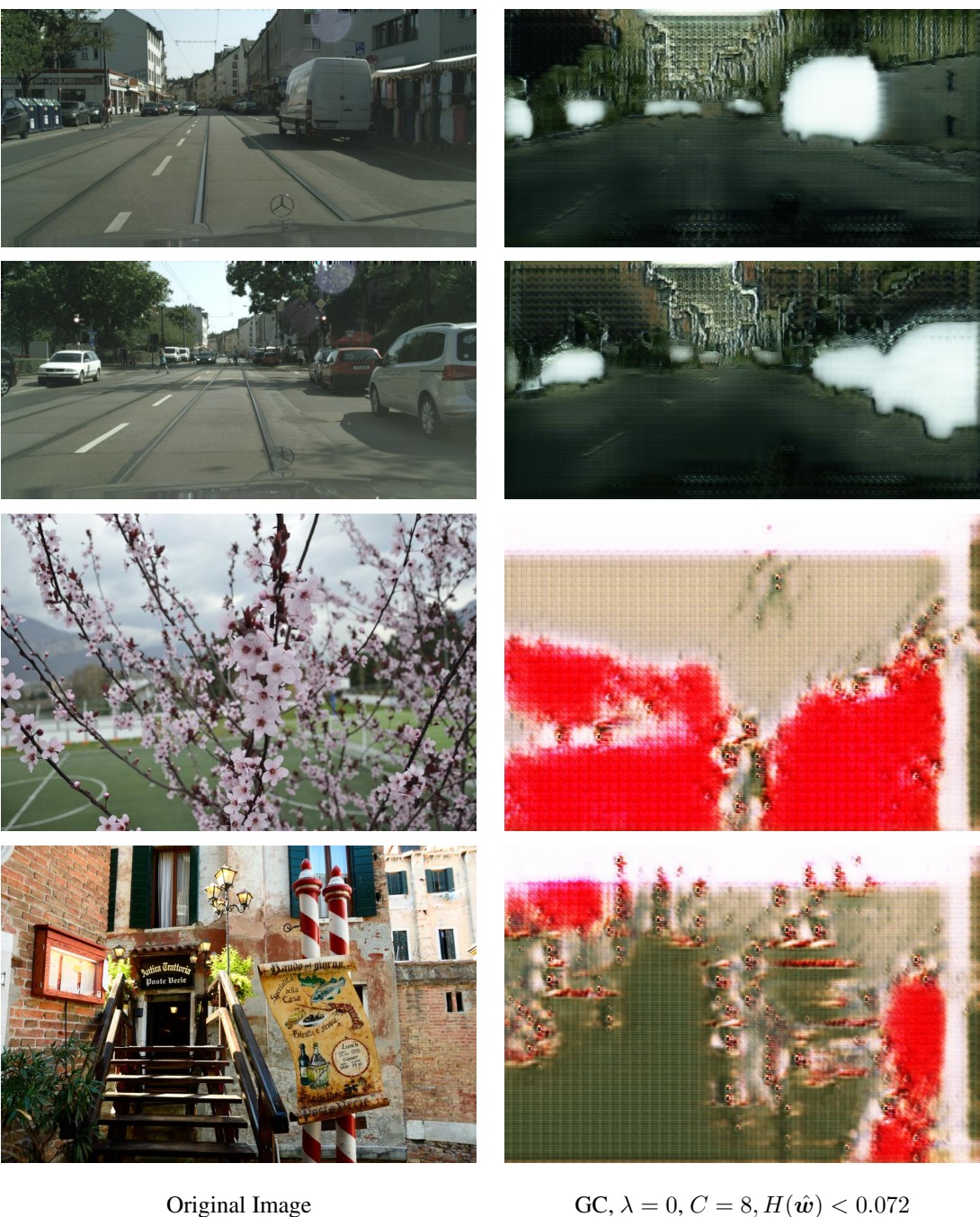

|                    |                                                    |
|--------------------|----------------------------------------------------|
| Original Image     | GC, $\lambda = 0$, $C = 8$, $H(\hat{w}) < 0.072$   |

Figure 25: When disabling the distortion losses (i.e. $\lambda = 0$), such that only $\mathcal{L}_{\text{GAN}}$ remains, we observe that the training "collapses" and produces repetitive textures, both for OpenImages and Citycapes.

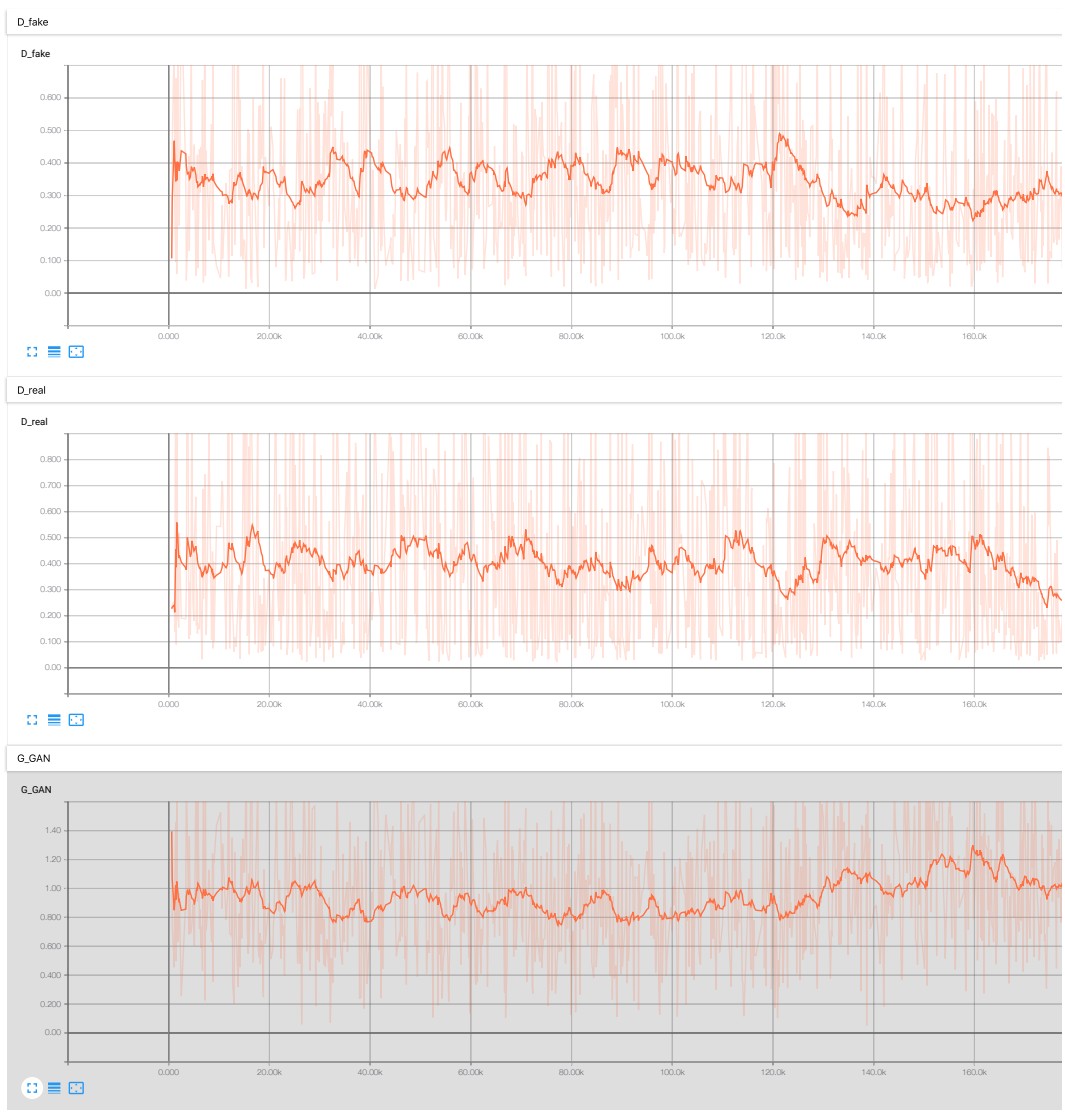

Figure 26: We show convergence plots for the generator and discriminator losses from training our GC $C = 8$) channel model on OpenImages.

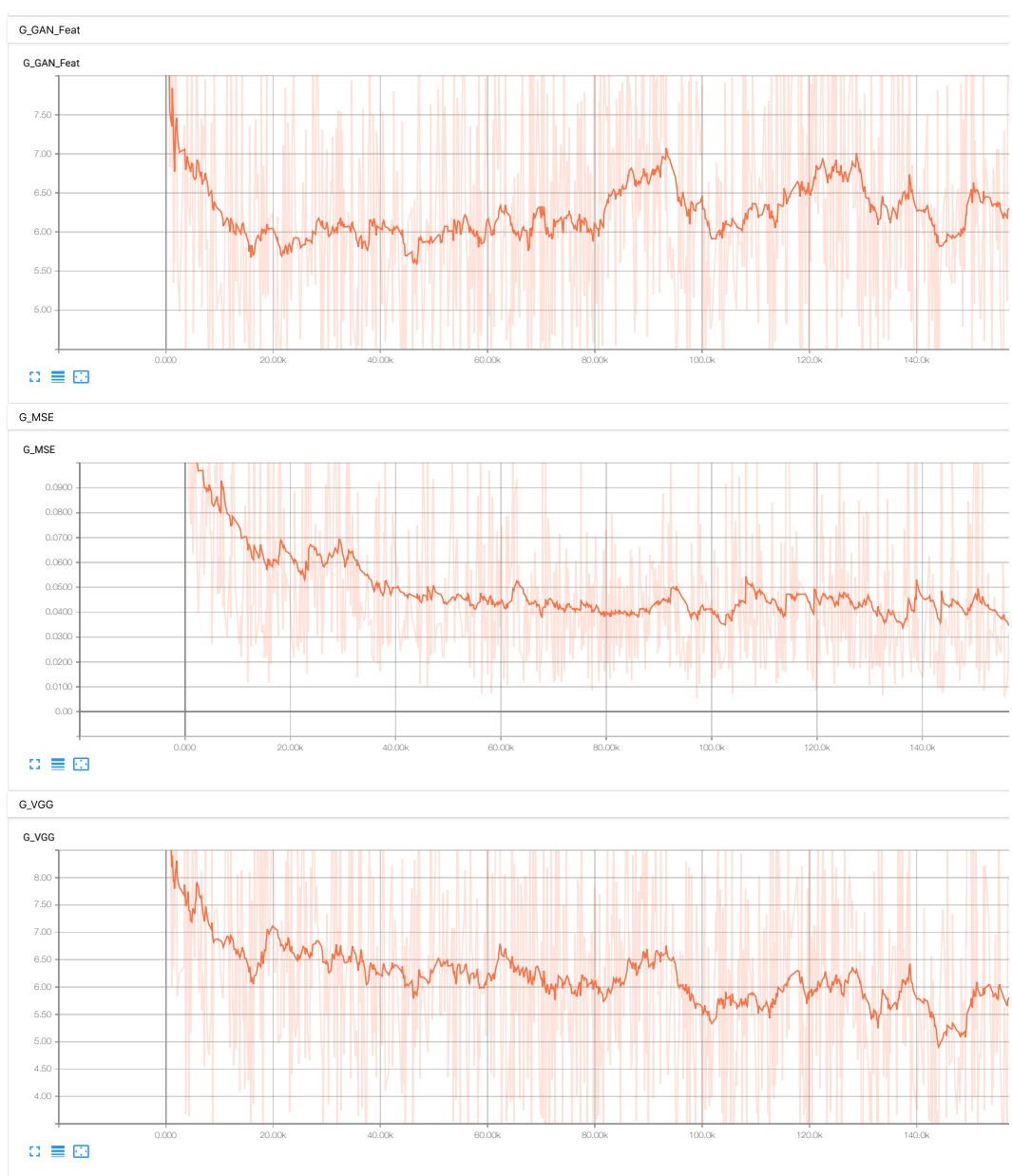

Figure 27: We show convergence plots for the distortion losses from training our GC $C = 8$) channel model on OpenImages.

