# OpenReview forum: "Generative Adversarial Networks for Extreme Learned Image Compression"
_ICLR.cc/2019/Conference_

### Official Review · AnonReviewer1 · 2018-11-02

**Rating:** 4
**Confidence:** 4

**Review:**

This paper proposes to use GAN to address the image compression problem. It is shown to achieve superior results over the past work in two different settings (GC and SC).

Novelty:

It has been well discovered in the literature of GANs that they can resolve the problem of blurriness in generation, compared to the traditional MSE loss. This paper proposes to combine a GAN loss with MSE, together with an entropy loss. However similar approaches were used such as video prediction [1] from 2016. The paper lacks a few references like this.

Major questions:

- How do the different loss terms play against each other? The entropy term and the MSE apparently conflict with each other. And how would this affect L_gan? I would like to request some more analysis of this or ablation study on different terms.

- How well does the GAN converge? A plot of G and D loss is often presented with GAN approaches.

- Discrete latent variable is in itself an interesting problem [2]. I see the image compression as a task to discover a discrete latent variable with minimal storage. Perhaps one most important problem is how to estimate the gradient through the discrete bottleneck. But the paper doesn't provide much insights or experiments on this.

- I'm not fully convinced by the claim of the noise that this paper uses to combine the code can act as a regularizer. Adding the noise makes the decoder output stochastic, but the compression problem seems to be deterministic by nature, unlike many other generation problems.

[1] https://arxiv.org/abs/1511.05440
[2] https://arxiv.org/abs/1711.00937

---

> ### Author Response · Authors · 2018-11-27
> **Reply to AnonReviewer1**
>
> We thank the reviewer for the feedback.
>
> Regarding novelty, we refer to the top comment  ( https://openreview.net/forum?id=HygtHnR5tQ&noteId=BJlLGD3c07 ). We will also add additional references on prior use of GANs for reducing blurriness as suggested.
>
> Regarding the interplay between losses: We note that the apparent conflict between the entropy term and MSE loss is motivated by rate-distortion theory (see Cover & Thomas, 2012, Chapter 13) and it is the standard approach to train compression networks in the current lossy compression literature (e.g. Balle et al., 2017, Theis et al., 2017, Minnen et al. (2018)). Note that in our approach, the entropy term is implicit in the dimensionality of the bottleneck and thereby only acts as an upper bound (see Eq. 5 in paper), implemented as a hard constraint rather than a regularizer (see below). It therefore does not conflict with the GAN loss, but rather provides the generator/decoder with conditional information. Regarding the interaction between the GAN loss and the MSE loss, we observe that the MSE loss stabilizes the training as it penalizes collapse of the GAN (see Fig 25, Appendix F.8, p. 29), and observe that the distortion (measured in PSNR) varies as expected when we vary the entropy constraint and turn off the GAN/distortion losses.
>
> Convergence Plots: we have added convergence plots in Figures 26&27 in Appendix F.8, p. 30-31. We note that the loss fluctuates heavily across iterations due to the small batch size (one), but the smoothed losses are stable. For all our experiments, both on Cityscapes and OpenImages, we kept the weights of the losses and ratio between discriminator/generator iterations constant and at point did our (GC and SC) models collapse during training for either dataset.
>
> Regarding gradients through the discrete bottleneck: We use the differentiable relaxation of the quantizer as proposed in (Mentzer et al., 2018, see p4), for which we omitted the details for brevity. Essentially, the “hard” (actual) quantization function argmax is replaced (for the backward pass only) with a “soft” quantization implemented with a softmax. For the forward pass, we use argmax, s.t. the decoder always receives quantized values. Other approaches from the learned compression literature use an approximation based on adding noise (Balle et al., 2016a, b) or using rounding for the forward pass and identity for the backward pass (Theis et al., 2017). It seems that all of these methods work reasonably well in the context of learned compression.
>
> We do visualize the learned discrete representation by sampling uniformly from the bottleneck and generating bottlenecks learned via WGAN-GP in Section 6.1. It can be seen in Figure 5 that uniform bottlenecks yield “soups of visual words”, but global coherence is lost. When the bottlenecks are generated via WGAN-GP, the global coherence becomes much better. This means that the quantized representation captures image content well beyond the pixel level.
>
> Regarding the regularizer: It seems that the reviewer has misread the statement about regularization: the reconstruction loss, not the noise acts as a regularizer.

---

### Official Review · AnonReviewer2 · 2018-11-04

**Rating:** 6
**Confidence:** 3

**Review:**

This paper proposed an interesting method using GANs for image compression. The experimental results on several benchmarks demonstrated the proposed method can significantly outperform baselines.

There are a few questions for the authors:

1.The actually benefit from GAN loss: the adversarial part usually can benefit the visual quality but is not necessary related to image quality (e.g. SSIM, PSNR).

2.The novelty of the model: GAN models with multiple G-Ds or local/global discriminators is not novel (see the references).

3.Do you have ablation study on the effects of conditional GAN and compression part to the model?

References:
a. Xi et al. Pedestrian-Synthesis-GAN: Generating Pedestrian Data in Real Scene and Beyond
b. Yixiao et al. FD-GAN: Pose-guided Feature Distilling GAN for Robust Person Re-identification

Revision: the rebuttal can not address my concerns, especially the image quality assessment and the novelty of the paper parts. I will keep my original score but not make strong recommendation to accept the paper.

---

> ### Author Response · Authors · 2018-11-27
> **Reply to AnonReviewer2**
>
> We thank the reviewer for the feedback.
>
> If you define the "image quality" as being SSIM/PSNR, obviously there is no benefit since that is not what we optimize for. However, we stress the contrast between the results in Figure 2 in the paper: while the PSNR optimized approaches have a higher "image quality" in terms of PSNR, they look much blurrier and have more artifacts.
> Motivated by this, we performed an extensive user study to confirm that our system results of higher visual quality, detailed in Section 6.1.
>
> Regarding novelty of GAN, we refer to the top comment ( https://openreview.net/forum?id=HygtHnR5tQ&noteId=BJlLGD3c07).&noteId=BJlLGD3c07 ). We will add further references for multiple G-Ds and local/global discriminators.
>
> Regarding ablation study: we refer to Figure 2. for a comparison between using GAN or MSE loss, as well as the user study on CityScapes (Fig. 4).  The user study furthermore shows that as the entropy constraint is varied (by increasing $C$), the visual quality improves. Additionally, in Table 2 (Appendix F.8, p. 28)  we consider the effect of varying the entropy constraint and the GAN/distortion losses for the PSNR on the Cityscapes test set (we stress again though that the PSNR on its own is not an indicator for the visual quality as seen in our user study, where we outperform state-of-the-art methods which have superior PSNR).

---

### Official Review · AnonReviewer3 · 2018-11-08
**Impressive results, but some details unclear**

**Rating:** 6
**Confidence:** 3

**Review:**

This paper proposed GAN-based framework for image compression and show improved results over several baseline methods. Although the approach is not very novel by itself, the adaption and combination of existing methods for the proposed solution is interesting. Although the bpp are consistently lower, the quality metrics used for comparison seem unclear.


Pros:
+ The reported compression results with a GAN-based framework for large images are impressive
+ Comprehensive set of results with Kodak, RAISE1K and Cityscapes datasets
+ The paper is well written with the core results and idea being well articulated


Cons:
+ Primary concern:  The quality metrics are unclear esp. for GC models, since traditional metrics such MS-SSIM and PSNR are noted to worse and primarily visual inspection is used for comparison, making it less concrete. Would also to help include these metrics for comparison
+ Eqn6: \lamda balancing the distortion between GAN loss and entropy terms - can the authors elaborate on this ? Furthermore, the ensuing statement that the addition of the distortion term, results in acting like a regularizer - seems like only a conjecture, can the authors additionally comment on this as well.


Minor issues:
+ The comparison of improvement in compression is reported using relative percentage numbers in some places as the improvement and others as lack of therein. It would help to use a common reporting notation throughout the text, this helps readability/understandability

---

> ### Author Response · Authors · 2018-11-27
> **Reply to AnonReviewer3**
>
> We thank the reviewer for the feedback.
>
> Please see the top level comment ( https://openreview.net/forum?id=HygtHnR5tQ&noteId=BJlLGD3c07 ) for general comments.
>
> Regarding the quality metrics: the MS-SSIM and PSNR are worse because of the GAN loss. When the network synthesizes texture, these details do not align with the original texture in the image, causing a higher PSNR than corresponding blurry regions from MS-SSIM/PSNR optimized models. For this reason, we conducted a thorough user study to assess the quality of our results in comparison with other methods, based on human perception (Sec 6.1 and Fig. 4).
>
> We stress the contrast in Figure 2 between our model trained with GAN and the MSE baseline using the same architecture. While the PSNR for the MSE baseline is more than 2dB higher than ours, the perceptual quality is clearly worse. In the additional results provided in Table 2 (Appendix F.8, p. 28) we show how the PSNR on Cityscapes vary as we vary the entropy constraint and the distortion/GAN losses.
>
> \lambda in Eq. (6) simply weights the distortion loss. If \lambda is very large, it dominates the total loss and the network will behave as a standard learned compression system. If it zero, then only the  GAN loss and the entropy loss/constraint remain. Similar to the challenges faced when training a standard GAN for high resolution images (the difference here is that we have access to quantized features from the encoder), we observe a collapse when turning off the distortion losses (Fig 25, Appendix F.8, p. 29). Without the distortion losses, the only signal for the generator/decoder is the "view of the discriminator" of the realism of the image, without any reference to the encoded input image. In contrast, when the distortion loss is added, the encoder and the decoder/generator get direct gradient information on how to to improve the output in reference to the input image.

---

### Author Response · Authors · 2018-11-27
**General Comment to all Reviewers**

All three reviewers are concerned with the novelty of using GANs for compression, so we address this here. We agree with the reviewers that multi-scale discriminators and the use of GANs to prevent blur have been considered before, and we do not claim that these on their own are novel contributions of ours.

However, we stress the following:

- Our approach for combining GANs with extreme compression has not been explored before. We focus on a new direction of image compression, where the algorithm focuses on realistic instead of faithful reconstructions w.r.t. PSNR/MS-SSIM, thereby obtaining mostly artifact-free reconstructions at extremely low bitrates. Notice how in Figure 1, all other approaches produce some sort of blocking artifacts or extreme blurring, while our approach synthesizes realistic tree textures. We believe that this is a valuable novel direction for the compression literature.

- No previous works thoroughly studied GANs in the context of full resolution image compression. We showed its effectiveness by setting new state-of-the-art in visual quality based on a user study (with dramatic bitrate savings), ablated on the differences between optimizing with a GAN loss vs MSE only (both visually (Fig. 3)  and in a user study (Fig. 4)) and show that the compressed representations learned are more meaningful than for MSE models when decoded (Fig. 20).

- No previous learned compression works explored these very low bitrates before for full resolution images. With our approach, we are the first to produce visually pleasing results at those bitrates.

---

> ### Author Response · Authors · 2018-12-01
> **Questions remaining**
>
> We hope the reviewers saw our rebuttal and we would be happy to answer any remaining questions.

---

### Meta-Review · Area_Chair1 · 2018-12-12
**Limited novelty**

**Confidence:** 4
**Recommendation:** Reject

**Metareview:**

This paper proposes a GAN-based framework for image compression.

The reviewers and AC note a critical limitation on novelty of the paper i.e., such a conditional GAN framework is now standard. The authors mentioned that they apply GAN for extreme compression for the first time in the literature, but this is not enough to justify the novelty issue.

AC thinks the proposed method has potential and is interesting, but decided that the authors need new ideas to publish the work.